# Early lock-in of structured and specialised information flows during neural development

**David P Shorten[1]\*, Viola Priesemann[2], Michael Wibral[3], Joseph T Lizier[1]**

[1]Centre for Complex Systems, Faculty of Engineering, The University of Sydney, Sydney, Australia; [2]Max Planck Institute for Dynamics and Self-Organization, Göttingen, Germany; [3]Campus Institute for Dynamics of Biological Networks, Georg August University, Göttingen, Germany

**Abstract** The brains of many organisms are capable of complicated distributed computation underpinned by a highly advanced information processing capacity. Although substantial progress has been made towards characterising the information flow component of this capacity in mature brains, there is a distinct lack of work characterising its emergence during neural development. This lack of progress has been largely driven by the lack of effective estimators of information processing operations for spiking data. Here, we leverage recent advances in this estimation task in order to quantify the changes in transfer entropy during development. We do so by studying the changes in the intrinsic dynamics of the spontaneous activity of developing dissociated neural cell cultures. We find that the quantity of information flowing across these networks undergoes a dramatic increase across development. Moreover, the spatial structure of these flows exhibits a tendency to lock-in at the point when they arise. We also characterise the flow of information during the crucial periods of population bursts. We find that, during these bursts, nodes tend to undertake specialised computational roles as either transmitters, mediators, or receivers of information, with these roles tending to align with their average spike ordering. Further, we find that these roles are regularly locked-in when the information flows are established. Finally, we compare these results to information flows in a model network developing according to a spike-timing-dependent plasticity learning rule. Similar temporal patterns in the development of information flows were observed in these networks, hinting at the broader generality of these phenomena.

**\*For correspondence:**
david.shorten@sydney.edu.au

**Competing interest:** The authors declare that no competing interests exist.

## Editor's evaluation

This work analyses how meaningful connections develop in the nervous system. The authors study the dissociated neuronal cultures and find that the information processing connections develop after 5–10 days. The direction of the information flow is influenced by neuronal bursting properties: the early bursting neurons emerge as sources and late bursting neurons become sinks in the information flow.

## Introduction

Throughout development, how do brains gain the ability to perform advanced computation? Given that the distributed computations carried out by brains require an intrinsic information processing capacity, it is of utmost importance to decipher the nature of the emergence of this capacity during development.

For brains to engage in the computations required for specific tasks, they require a general-purpose computational *capacity*. This capacity is often studied within the framework of information dynamics, where it is decomposed into the atomic operations of information storage, transfer, and modification (*Lizier et al., 2014*; *Lizier, 2013*). We are particularly interested in the information flow component, which is measured using the transfer entropy (TE) (*Schreiber, 2000*; *Bossomaier et al., 2016*). There exists a substantial body of work examining the structure and role of computational capacity in terms of these operations in mature brains. This includes the complex, dynamic, structure of information transfer revealed by calcium imaging (*Orlandi et al., 2014*), fMRI (*Mäki-Marttunen et al., 2013*; *Lizier et al., 2011*), MEG (*Wibral et al., 2011*), and EEG (*Shovon et al., 2016*; *Huang et al., 2015*; *Stramaglia et al., 2012*; *Marinazzo et al., 2014a*), and the role of information storage in representing visual stimuli (*Wibral et al., 2014a*), among others.

Given the established role of information flows in enabling the computations carried out by mature brains, we aim to study how they self-organise during neural development. There are a number of requirements for such a study. Firstly, it needs to be performed at a fine spatial scale (close to the order of individual neurons) to capture the details of development. It also needs to be conducted longitudinally in order to track changes over developmental timescales. Finally, the estimation of the information flow as measured by TE needs to be performed with a technique which is both accurate and able to capture the subtleties of computations performed on both fine and large timescales simultaneously.

Considering the first requirement of fine spatial scale, cell cultures plated over multi-electrode arrays (MEAs) allow us to record from individual neurons in a single network, providing us with this fine spatial resolution. There have been a number of previous studies examining information flows in neural cell cultures, for example, *Nigam et al., 2016*; *Shimono and Beggs, 2015*; *Matsuda et al., 2013*; *Timme et al., 2014*; *Kajiwara et al., 2021*; *Timme et al., 2016*; *Wibral et al., 2017*. Such work has focussed on the directed functional networks implied by the estimated TE values between pairs of nodes, which has revealed interesting features of the information flow structure. See section 'Previous application of the discrete-time estimator' for a more detailed description of this previous work.

However, moving to our second requirement of a longitudinal study, these studies have almost exclusively examined only single points in neural development since nearly all of them examined recordings from slice cultures of mature networks. By contrast, we aim to study the information flows longitudinally by estimating them at different stages in development. Using recordings from developing cultures of dissociated neurons (*Wagenaar et al., 2006b*) makes this possible.

In terms of our third requirement of accurate and high-fidelity estimation of TE, we note that all previous studies of information flows in neural cell cultures made use of the traditional discrete-time estimator of TE. As recently demonstrated (*Shorten et al., 2021*), the use of this estimator is problematic as it can only capture effects occurring on a single timescale. In contrast, a novel continuous-time TE estimator (*Shorten et al., 2021*) captures effects on multiple scales, avoiding time binning, is data efficient, and consistent. See section 'Transfer entropy estimation' for a more detailed discussion of the differences between the continuous-time and discrete-time estimators.

In this article, we thus examine the development of neural information flows for the first time, addressing the above requirements by applying the continuous-time TE estimator to recordings of developing dissociated cultures. We find that the amount of information flowing over these cultures undergoes a dramatic increase throughout development and that the patterns of these flows tend to be established when the flows arise. During bursting periods, we find that nodes often engage in specialised computational roles as either transmitters, receivers, or mediators of information flow. Moreover, these roles usually correspond with the node's mean position in the burst propagation, with middle bursters tending to be information mediators. This provides positive evidence for the pre-existing conjecture that nodes in the middle of the burst propagation play the vital computational role of 'brokers of neuronal communication' (*Schroeter et al., 2015*). Intriguingly, the designation of computational roles (transmitter, receiver, or mediator) appears to also be determined early when the information flows are established. Finally, in order to investigate the generality of these phenomena, as well as a putative mechanism for their emergence, we study the dynamics of information flow in a model network developing according to a spike-timing-dependent plasticity (STDP) (*Caporale and Dan, 2008*) update rule. We find that the abovementioned phenomena are present in this model system, hinting at the broader generality of such patterns of information flow in neural development.

## Results

Data from overnight recordings of developing cultures of dissociated cortical rat neurons at various stages of development (designated by days in vitro [DIV]) was analysed. These recordings are part of an open, freely available, dataset (*Wagenaar et al., 2006b*; *Network, 2021*). See 'Materials and methods' (section 'Cell culture data') for a summary of the setup that produced the recordings. We studied all cultures for which there were overnight recordings. We restricted our analysis to these overnight recordings as they provided sufficient data for the estimation of TE. In what follows, we refer to the cultures by the same naming convention used in the open dataset: 1-1 through 1-5 and 2-1 through 2-6. The majority of cultures have recordings at three different time points, three have recordings at four points (1-3, 2-2, and 2-5) and one has only two recordings (2-1). The days on which these recordings took place vary between the 4th and 33rd DIV. By contrasting the TE values estimated at these different recording days, we are able to obtain snapshots of the emergence of these culture's computational capacity.

In the analysis that follows, for space considerations, we show plots for four representative cultures: 1-1, 1-3, 2-2, and 2-5. The latter three were chosen as they were the only cultures with four recording days. Culture 1-1 was selected from the group with three recording days, having the latest final recording day that was no more than a week later than the penultimate recording day. Plots for the remaining cultures are shown in Appendix 3. We also display certain summary statistics for the results of all cultures in the main text. Culture 1-5 is anomalous (among the recordings studied in this work) in that on its final day it has ceased to burst regularly, as stated in Figure 3A of *Wagenaar et al., 2006b*. This leads to its results being substantially different from those of the other cultures, as will be presented below.

The TE between all pairs of electrodes was estimated using a recently introduced continuous-time estimator (*Shorten et al., 2021*; see section 'Transfer entropy estimation'). This produces a directed functional network at each recording day, and we aim to analyse how the connections in this network change over development time. Spike sorting was not performed because we would not be able to match the resulting neural units across different recordings and could not then fulfil our aim of contrasting the information flow between specific source-target pairs at different recording days. As such, the activity on each node in the directed functional networks we study is multi-unit activity (MUA) (*Schroeter et al., 2015*) formed of the spikes from all neurons detected by a given electrode, with connections representing information flows in the MUA. For more details on data preprocessing as well as the parameters used with the estimator, see 'Materials and methods'.

### The dramatic increase in the flow of information during development

We first investigate how the amount of information flowing between the nodes changes over the lifespan of the cultures. *Table 1* shows the mean TE between all source-target pairs (*Appendix 3—table 1* shows these values for the additional cultures). We observe that this mean value increases monotonically with the number of DIV, with only a single exception in the main cultures (a slight drop in the mean TE between days 21 and 33 of culture 2-2). We can make the same observation for the additional cultures, where the only drop is caused by day 5 of culture 2-4, a day still very early in development which had no significant TE values. We performed a two-sided Student's *t*-test

**Table 1.** Mean transfer entropy (TE) in nats per second between every source-target pair for each recording studied.

| Culture 1-1 | Day 4 | Day 14 | Day 20 | |
|---|---|---|---|---|
| | 0 | 0.060 | 0.097 | |
| Culture 1-3 | Day 5 | Day 10 | Day 16 | Day 24 |
| | 0 | $2\times10^{-4}$ | 0.017 | 0.098 |
| Culture 2-2 | Day 9 | Day 15 | Day 21 | Day 33 |
| | 0 | 0.015 | 0.11 | 0.057 |
| Culture 2-5 | Day 4 | Day 10 | Day 22 | Day 28 |
| | 0 | $2\times10^{-3}$ | 0.037 | 0.082 |

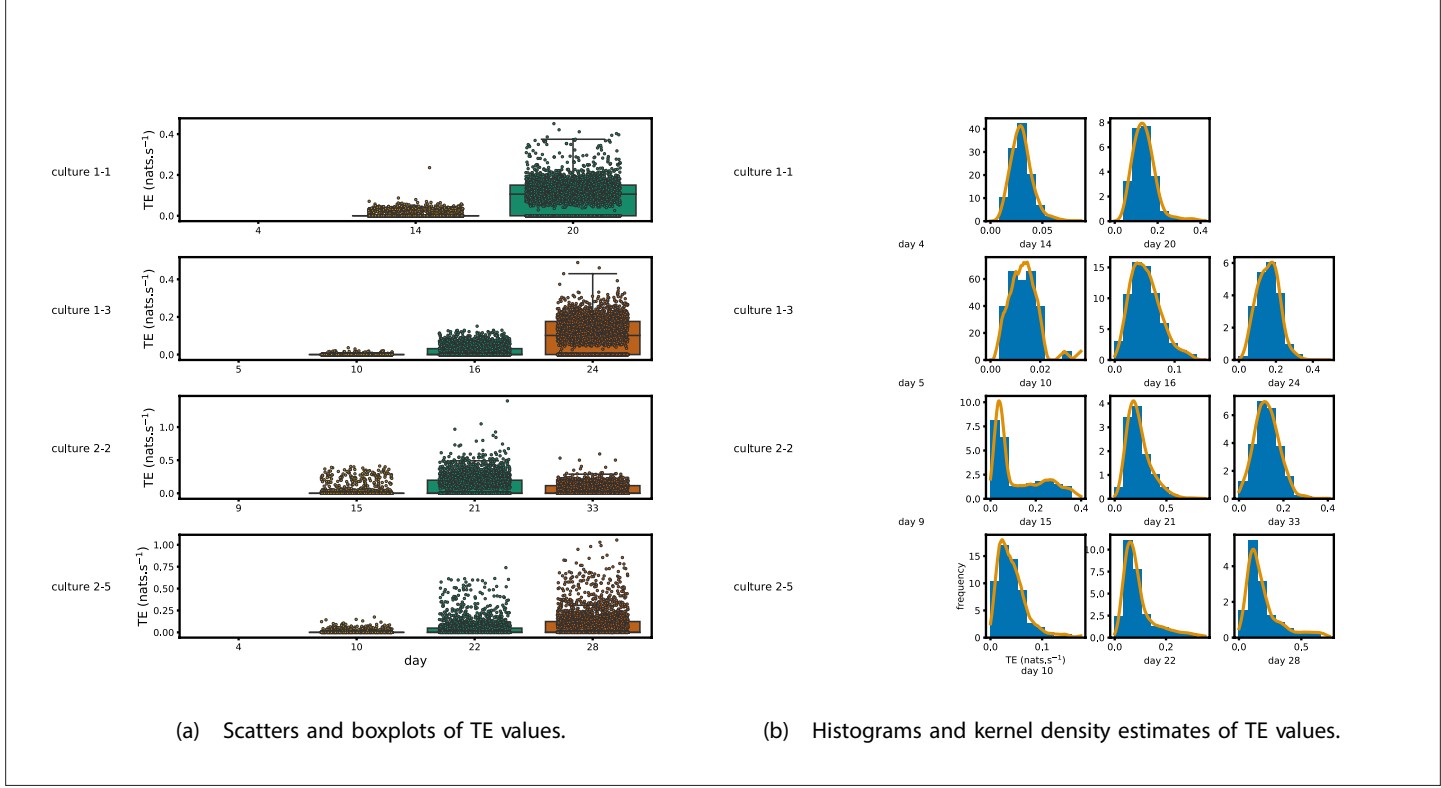

(a)  Scatters and boxplots of TE values.  (b)  Histograms and kernel density estimates of TE values.

**Figure 1.** Plots of the distributions of estimated transfer entropy (TE) values in the recordings analysed in this study. (**a**) Scatters of the TE values are overlaid on box plots. The box plots show the quartiles and the median (values greater than 10 SDs from the mean have been removed from both the box and scatter plots as outliers). (**b**) Density estimates of the nonzero (statistically significant) TE distribution on top of a histogram. The densities are estimated using a Gaussian kernel. The histogram bin width and kernel histogram are both 10% of the data range. Recordings with fewer than 10 statistically significant TE values are excluded.

for the difference in the mean between all pairs of recordings for each culture. All such differences (increases and decreases) were found to be statistically significant at $p<0.01$ with Bonferroni correction for multiple comparisons.

Overall, the magnitude of the increase in the mean TE is substantial. All the first recordings for the main cultures had a mean estimated TE of $0\,\mathrm{nats.s}^{-1}$ (with no statistically significant transfer entropies measured as per section 'The emergence of functional information flow networks'). By contrast, all recordings beyond 20 DIV had a mean TE greater than $0.037\,\mathrm{nats.s}^{-1}$.

*Figure 1* shows scatter plots of the TE values in each recording laid over box-and-whisker plots (*Appendix 3—figure 1* shows equivalent plots for the additional cultures). The large increase over time in the amount of information flowing over the networks is clearly visible in these plots. However, it is interesting to note that certain source-target pairs do have large information flows between them on early recording days even while the average remains very low.

*Figure 1b* shows histograms of the TE values estimated in each recording along with probability densities estimated using a Gaussian kernel (*Appendix 3—figure 1* shows these for the additional runs). The distributions only include the nonzero (statistically significant) estimated TE values. Some of these distributions do, qualitatively, appear to be log-normal, in particular for later DIV. Moreover, previous studies have placed an emphasis on the observation of log-normal distributions of TE values in in vitro cultures of neurons (*Shimono and Beggs, 2015*; *Nigam et al., 2016*). As such, we quantitatively analysed the distribution of the nonzero (statistically significant) estimated TE values in each individual recording. However, contrary to expectations, we found that these values were not well described by a log-normal distribution. It is worth noting that previous work which analysed the distribution of TE values in networks of spiking neurons was performed on organotypic cultures, as opposed to the dissociated cultures studied in this work. See Appendix 1 for further details and discussion.

## The emergence of functional information flow networks

By considering each electrode as a node in a network, we can construct directed functional networks of information flow by assigning a directed edge between each source-target pair of electrodes with a statistically significant information flow. This results in weighted directed networks, the weight being provided by the TE value. Diagrams of these networks for the main cultures are shown in *Figure 2*, and in *Appendix 3—figure 3* for the additional cultures. Note that, in all subsequent analysis presented in this article, a TE value of zero was assigned to all cases where the TE was not statistically significant.

We are able to notice a number of interesting spatiotemporal patterns in these diagrams. Firstly, the density (number of edges) of the networks increases over time. This is quantified in *Table 2*, which shows the number of source-target pairs of electrodes for which a statistically significant nonzero TE value was estimated. In all the main cultures studied, the number of such pairs (and, therefore, the network density) increased by orders of magnitude over the life of the culture. For instance, in all four cultures, no statistically significant TE values are estimated on the first recording day. However, over 1000 source-target pairs have significant TE values between them on the final day of recording for each culture. *Appendix 3—table 2* shows the same values for the additional cultures. With a few exceptions (such as the abovementioned anomalous culture 1-5), the same relationship is observed. Note that the final recording day for culture 2-4 is relatively early (day 11), and so the low number of significant edges on this day is consistent with the other cultures.

We are, therefore, observing the networks moving from a state where no nodes are exchanging information, to one in which information is being transferred between a substantial proportion of the pairs of nodes (≈30% density of possible directed connections in most networks). Put another way, the functional networks are emerging from an unconnected state to a highly connected state containing the information flow structure that underpins the computational capacity of the network. This helps to explain the overall increase in information flow across the network reported in section 'The dramatic increase in the flow of information during development'.

We observe that the information flow (both incoming and outgoing) is spread somewhat evenly over the networks – in the sense that in the later, highly connected, recordings there are very few areas with neither incoming nor outgoing flow (the one notable exception to this is culture 2-6). A number of clear hubs (with particularly high either outgoing or incoming information flow) do stand out against this strong background information flow however. The strongest such hubs (with many high-TE edges) are all information sinks: they have low outgoing information flow, but receive high flow from a number of other nodes.

One can observe many instances in these diagrams where nodes have either very high incoming flow and very low outgoing flow, or very low incoming flow and very high outgoing flow. That is, they are taking on the roles of source (information-transmitting) hubs or target (information-receiving) hubs. Notable instances of information-receiving hubs include node 49 of day 16 of culture 1-3, node 42 of day 22 of culture 2-5, and node 5 of day 15 of culture 2-2 (see *Figure 2* for the node numbers used here). Notable examples of information transmitting hubs include node 28 of day 10 of culture 1-3 and nodes 18, 19, 22, and 30 of day 22 of culture 2-5. The specialist computational roles that nodes can take on will be studied in more detail quantitatively in section 'Information flows quantify computational role of burst position', with a particular focus on how this relates to the burst propagation.

It is possible to observe some notable instances whereby the information processing properties of a node remain remarkably similar across recording days. For example, nodes 55, 50, and 39 of culture 2-2 are outgoing hubs (with almost no incoming TE) on all four recording days. This offers us a tantalising hint that the information processing structure of these networks might be locked-in early in development, being reinforced as time progresses. Section 'Early lock-in of information flows' performs a quantitative analysis of this hypothesis.

## Early lock-in of information flows

In the previous subsection, analysis of the directed functional networks of information flow suggested that the structure of the information processing capacity of the developing networks might be determined early in development and reinforced during the subsequent neuronal maturation.

In order to quantitatively investigate this hypothesis, we examine the relationships in the information flow from a given source to a given target between different recording days. That is, we are probing whether the amount of information flowing between a source and a target on an early day

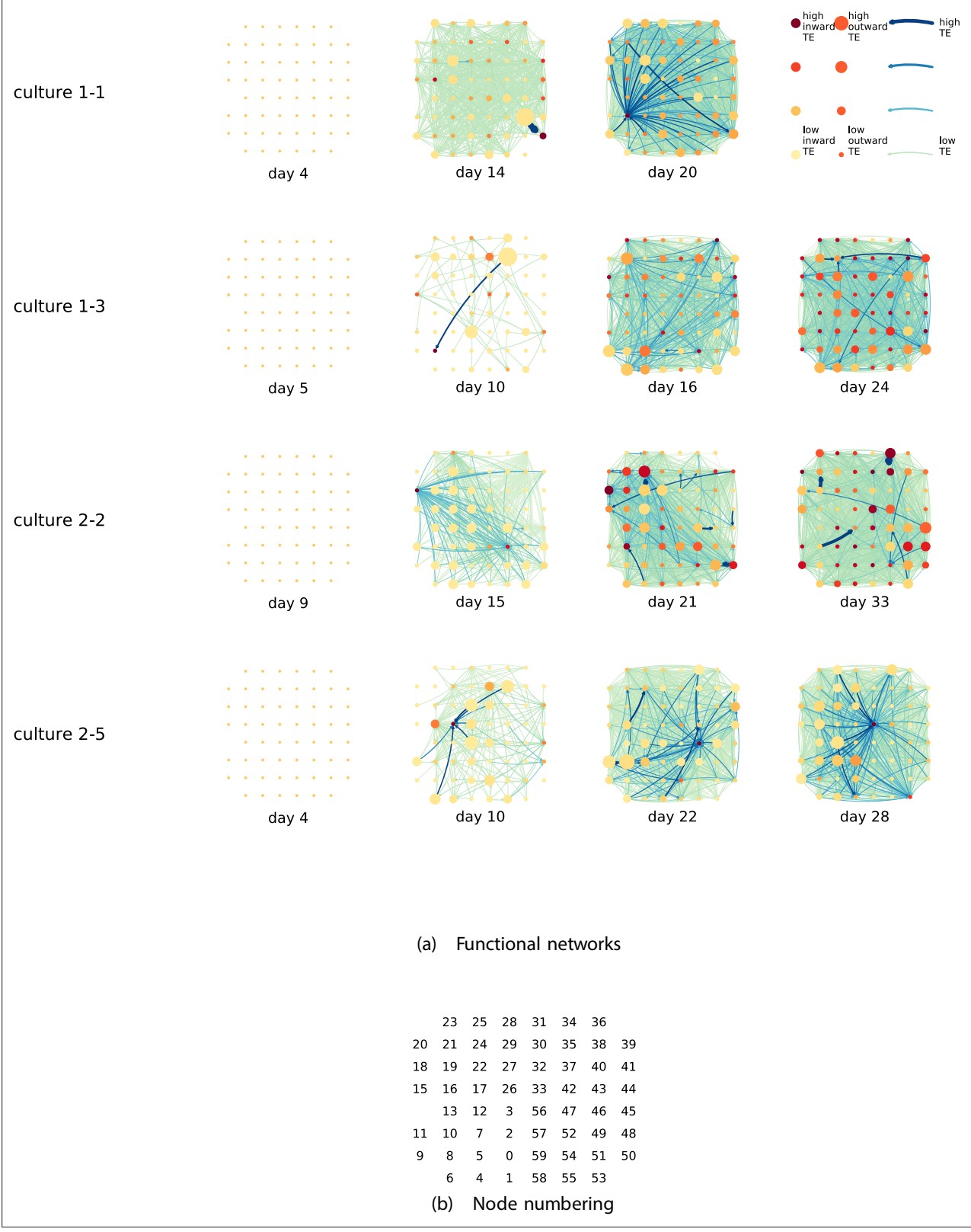

**Figure 2.** Functional networks overlaid on the spatial layout of the electrodes. (**a**) The directed functional networks implied by the estimated transfer entropy (TE) values. Each node represents an electrode in the original experimental setup. The nodes are spatially laid out according to their position in the recording array. An edge is present between nodes if there is a statistically significant information flow between them. The edge weight and colour are indicative of the amount of information flowing between electrodes (see the legend). The scaling of this weight and colour is done relative to the

*Figure 2 continued on next page*

*Figure 2 continued*

mean and variance of the information flow in each recording separately. The size and colour of the nodes are assigned relative to the total outgoing and incoming information flow on the node, respectively. As with the edge colour and size, this is done relative to the distribution of these values in each recording separately. (**b**) The spatial layout of the nodes. The numbering is identical to that used in the documentation of the open dataset studied in this work (*Wagenaar et al., 2006b*; *Network, 2021*).

of development will be correlated with the amount flowing on a later day of development. This is equivalent to studying the correlation in the weights of the edges of the functional networks across different recording days. *Figure 3* shows scatter plots between the TE values estimated between each source-target pair on earlier and later days. Note that, in every case where the null hypothesis of zero TE could not be rejected (the TE was not statistically significant), a value of zero was used. Days with fewer than 10 nonzero values were excluded from the analysis as they could not lead to meaningful insights. By observing the pair scatters in *Figure 3a–d* (equivalent plots for the additional cultures are shown in *Appendix 3—figure 4*), we see that, in many pairs of days, there appears to be a substantial correlation between the TE values on the edges across days. This is particularly pronounced for cultures 1-3 and 1-1, though visual assessment of the trend is complicated by the many zero values (where TE is not significant), gaps in the distribution and outliers. As such, *Figure 3a–d* also display the Spearman rank-order correlation ($\rho$) for each early-late pair of days for each culture. This correlation is positive and statistically significant at the p<0.01 level (after Bonferroni correction for multiple comparisons) between all the final pairs of recording days in the analysed cultures (including those in the additional cultures). *Table 3* summarises the proportions of pairs of recordings (including the additional cultures) which had significant positive Spearman correlations between the TE values on the edges across days. Whether we focus on either the final pairs of recordings or also include pairs that occur after day 15 (by which time the information flow networks have emerged), either all or all but one of the pairs of recordings exhibit such correlations. Moreover, the probability of this number of correlations arising by chance is found to be very low. This represents a strong tendency for the relatively strong information flows between a given source and target on later days to be associated with the relatively strong information flow between the same source and target on an earlier day of development. *Figure 3* displays these Spearman correlations between the early and late TE between source-target pairs visually. We notice a trend, whereby the correlation of the TE values seems to be higher between closer days (sample point being closer to the diagonal) and where those days are later in the development of the cultures (sample points being further to the right).

We also investigated the manner in which a node's tendency to be an information source hub might be bound once information flows are established. *Figure 4* shows scatter plots between the outgoing TE of each node (averaged across all targets) on different days of development along with the associated Spearman correlations (*Appendix 3—figure 5* shows these plots for the additional cultures). By observing the scatter plots, it is easy to see that there is a strong positive relationship between the outgoing information flow from a given node on an earlier day of development and the outgoing

**Table 2.** The number of source-target pairs of electrodes with a statistically significant transfer entropy (TE) value between them for each recording studied.

This corresponds to the number of possible edges in the functional networks shown in *Figure 2*. As the electrode arrays used to record the data had 59 electrodes, the total number of unique ordered pairs of electrodes (and, therefore, the number of possible edges) is 3422.

| Culture 1-1 | Day 4 | Day 14 | Day 20 | |
|---|---|---|---|---|
| | 0 | 607 | 2166 | |
| Culture 1-3 | Day 5 | Day 10 | Day 16 | Day 24 |
| | 0 | 44 | 999 | 1902 |
| Culture 2-2 | Day 9 | Day 15 | Day 21 | Day 33 |
| | 0 | 371 | 1409 | 1386 |
| Culture 2-5 | Day 4 | Day 10 | Day 22 | Day 28 |
| | 0 | 185 | 975 | 1263 |

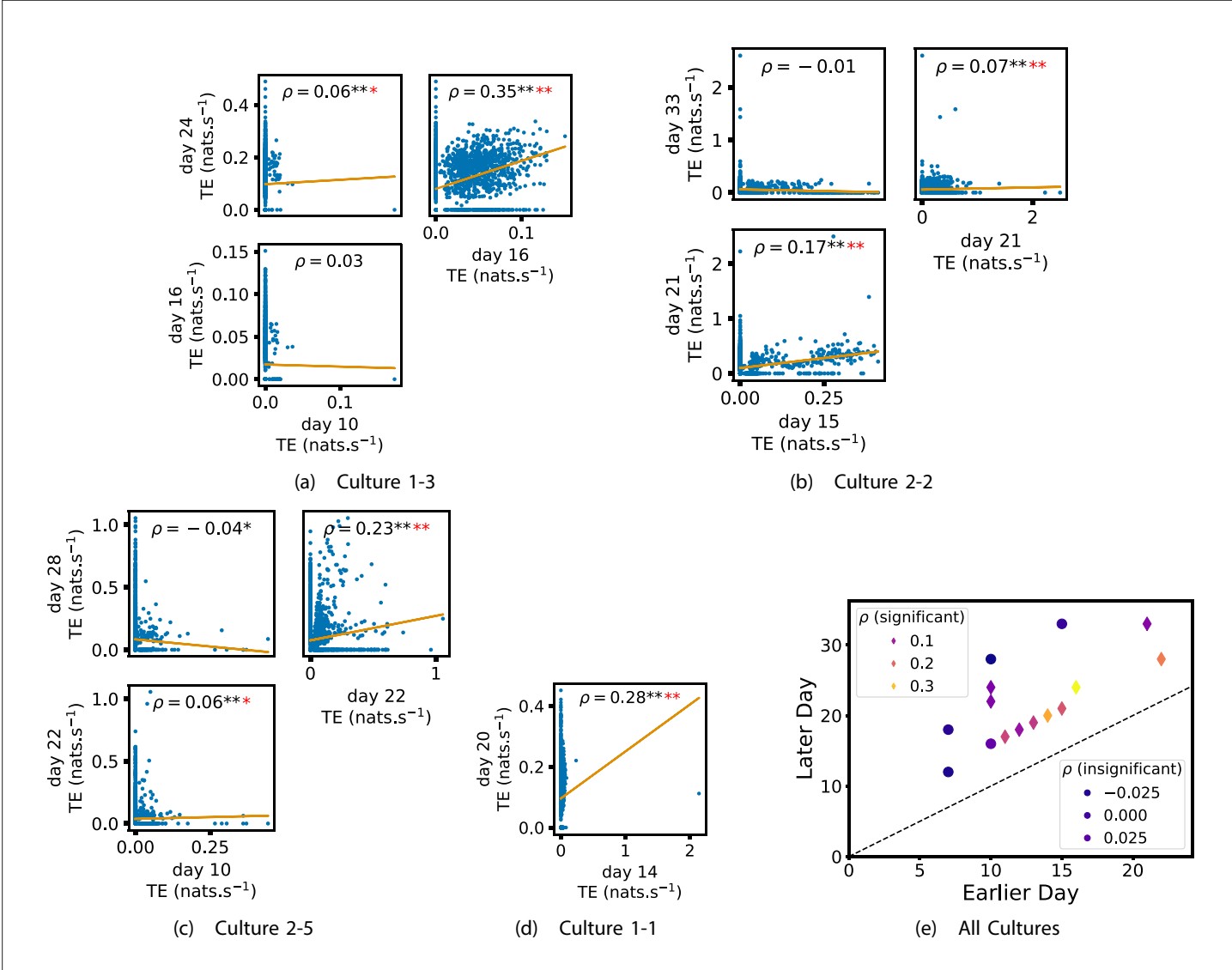

**Figure 3.** Plots investigating the relationship between the information flow on a given source-target pair over different days of development. (**a**–**d**) show scatter plots between all pairs of days for each culture (excluding days with less than 10 significant transfer entropy [TE] values). Specifically, in each scatter plot, the $x$ value of a given point is the TE on the associated edge on an earlier day and the $y$ value of that same point is the TE on the same edge but on a later day. The days in question are shown on the bottom and sides of the grids of scatter plots. The orange line shows the ordinary least-squares regression. The Spearman correlation ($\rho$) between the TE values on the two days is displayed in each plot. Values of $\rho$ significant at the 0.05 level are designated with an asterisk and those significant at the 0.01 level are designated with a double asterisk. Red asterisks are used to denote significance after performing a Bonferroni correction for multiple comparisons. (**e**) shows all recording day pairs for all cultures (where the pairs are always from the same culture) and the associated Spearman correlation between the TE on the edges across this pair of recording days. Diamonds indicate significance at p<0.05, with Bonferroni correction.

flow from that same node on a later day (when we restrict ourselves to focusing on pairs occurring after a substantial number of statistically significant information flows have been established). This is not surprising, given the correlation we already established for TE on individual pairs, but does not automatically follow from that. As with the TE on the edges, *Table 3* summarises the proportions of pairs of recordings (including the additional cultures) which had significant positive Spearman correlations between the outgoing TE from each node across days. We see that, whether we focus on just the last recordings of each culture or also include those after day 15, a substantial majority of pairs of recordings exhibit such correlations and that such a majority would be very unlikely to arise by chance. Some of these correlations are particularly strong, and indeed stronger than that observed on the TEs

**Table 3.** Summary of significance values for the lock-in results.
Each table cell shows the number of relationships that were found to be significant at the $p<0.05$ level, out of the total number of relationships tested. Note that relationships were only tested in cases where both recordings in the pair had at least 10 significant transfer entropy (TE) values. A hypothesis test is conducted against the null hypothesis that the original p-values that produced these results were uniformly distributed between 0 and 1 (giving a 0.05 chance of a significant result). * indicates that this probability is less than 0.05. ** indicates that the probability of the observed number of significant results has probability less than 0.001 under this null hypothesis, with a Bonferroni correction for multiple comparisons. The first row summarises the significance values for the value of the TE on the edges, shown in *Figure 3* and *Appendix 3—figure 4*. The second and third rows summarise the significance values for the mean outward and inward TE on each node, shown in *Figure 4* and *Appendix 3—figure 5* as well as *Appendix 2—figure 1* and *Appendix 3—figure 6*, respectively. The final row summarises the significance values for the ratio of inward to outward burst-local TE, shown in *Figure 6* and *Appendix 3—figure 9*. The columns which restrict the analysis to bursting recordings exclude the final recording of culture 1-5, as this culture ceased bursting, after having previously been bursty (*Wagenaar et al., 2006b*).

| | All cultures, final pair | All cultures, final pair or post day 15 | Bursting, final pair | Bursting, final pair or post day 15 |
|---|---|---|---|---|
| Edge | 7/7** | 8/9** | 6/6** | 7/8** |
| Out | 5/7** | 6/9** | 5/6** | 6/8** |
| In | 3/7* | 3/9* | 3/6* | 3/8* |
| Burst-local ratios | 5/7** | 6/9** | 5/6** | 6/8** |

of individual node pairs. For instance, between days 16 and 24 of culture 1-3 we have that $\rho = 0.62$ and between days 14 and 20 of culture 1-1 we have that $\rho = 0.71$. *Figure 4* visualises all Spearman correlations between the early and late total outgoing TE of a given node. As per the TEs for individual node pairs, the correlation is higher between closer days and where those days are later in the development of the cultures.

*Appendix 2—figure 1* shows similar plots to *Figure 4*, but for the average inward TE on each node (with *Appendix 3—figure 6* showing plots for the additional cultures). We observe three cases of significant correlations in this value between early and late days, indicating a weaker yet still statistically significant (*Table 3*) propensity for the average inward TE to also lock-in.

Of course, some pairs involving earlier days of some cultures (such as day 10 of cultures 1-3 and 2-5) do not exhibit such lock-in tendencies. However, as displayed in *Table 2*, there are very few significant information flows at this early stage of development (44 and 185, respectively). This represents a point in development perhaps too early for any substantial information flow networks to have emerged.

In summary, the data suggests that, in these developing neural cell cultures, the structure of the information flows is to a large degree locked-in early in development, around the point at which the information dynamics emerge. There is a strong tendency for properties of these flows on later days to be correlated with those same properties on earlier days. Specifically, we have looked at the flows between source-target pairs, the average outgoing flow from a source, and the average incoming flow to a target. The values of these variables on later DIV were found, in the majority of cases, to be positively correlated with the same values on earlier DIV. Further, there were no cases where a statistically significant negative correlation was found.

## Information flows quantify computational role of burst position

Developing cultures of dissociated neurons have a tendency to self-organise so as to produce population bursts or avalanches (*Wagenaar et al., 2006b*; *Pasquale et al., 2008*). Such spike avalanches are not only a feature of cell cultures, being a ubiquitous feature of in vivo neural recordings (*Priesemann et al., 2014*; *Priesemann et al., 2013*; *Priesemann et al., 2009*). There is a wide body of work discussing the potential computational importance of such periods of neuronal activity (*Lisman, 1997*; *Krahe and Gabbiani, 2004*; *Shew et al., 2011*; *Kinouchi and Copelli, 2006*; *Haldeman and Beggs, 2005*; *Rubinov et al., 2011*; *Cramer et al., 2020*). It has been observed that cultures often follow one

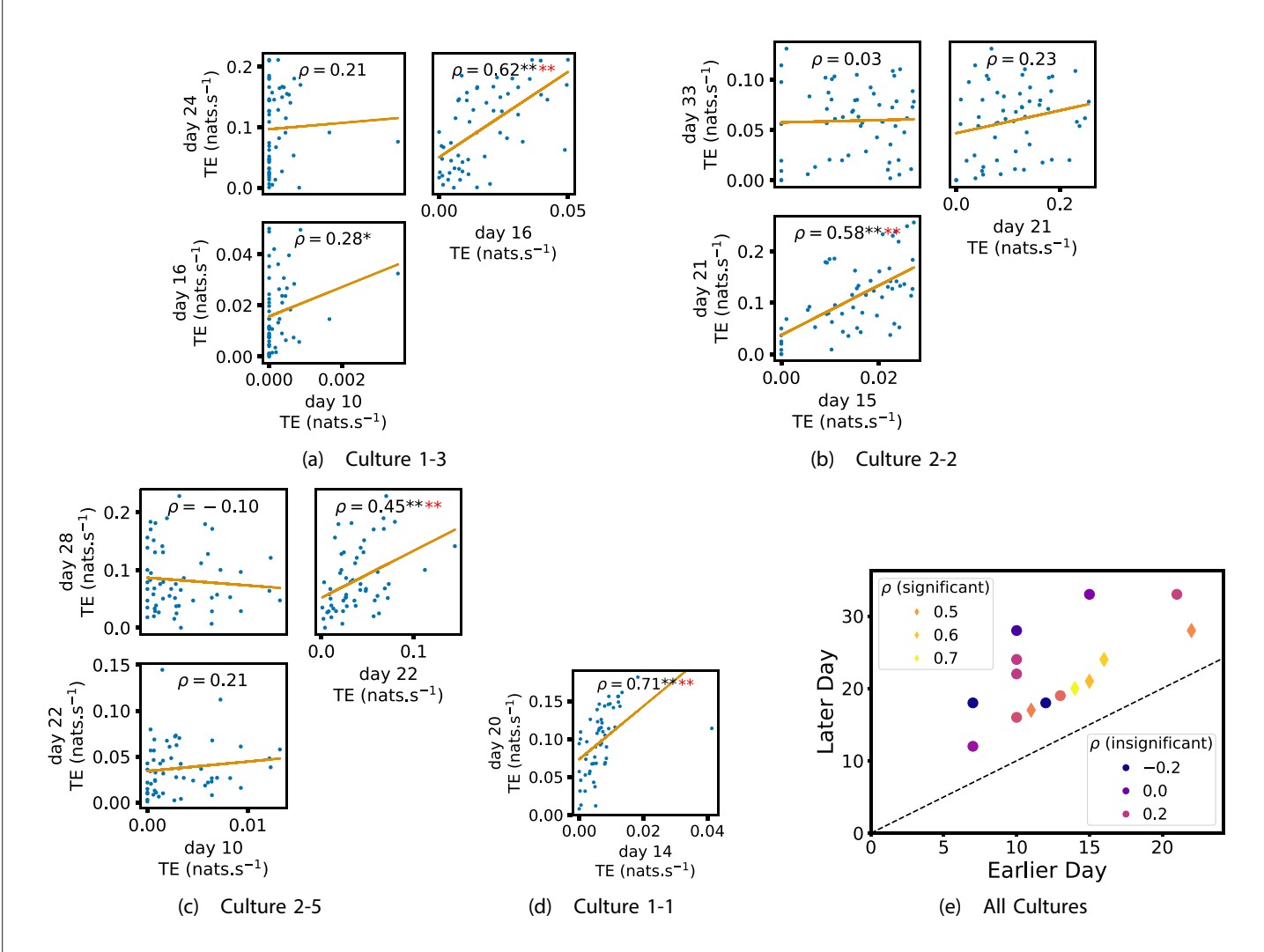

**Figure 4.** Plots investigating the relationship between the outward information flow from a given node over different days of development. (**a–d**) show scatter plots between all pairs of days for each culture (excluding days with less than 10 significant transfer entropy [TE] values). Specifically, in each scatter plot, the $x$ value of a given point is the average outgoing TE from the associated node on an earlier day and the $y$ value of that same point is the total outgoing TE from the same node but on a later day. The days in question are shown on the bottom and sides of the grids of scatter plots. The orange line shows the ordinary least-squares regression. The Spearman correlation ($\rho$) between the outgoing TE values on the two days is displayed in each plot. Values of $\rho$ significant at the 0.05 level are designated with an asterisk and those significant at the 0.01 level are designated with a double asterisk. Red asterisks are used to denote significance after performing a Bonferroni correction for multiple comparisons. (**e**) shows all recording day pairs for all cultures (where the pairs are always from the same culture) and the associated Spearman correlation between the outward TEs of nodes across this pair of recording days. Diamonds indicate significance at p<0.05, with Bonferroni correction.

or more ordered patterns of burst propagation (*Maeda et al., 1995*), with some nodes exhibiting a tendency to burst towards the beginning of these patterns and others towards their end (*Schroeter et al., 2015*). More recent work has proposed that the nodes which tend to burst at different points in these progressions play different computational roles (*Schroeter et al., 2015*). This work has placed special importance on those nodes which usually burst during the middle of the burst progression, conjecturing that they act as the 'brokers of neuronal communication'.

The framework of information dynamics is uniquely poised to illuminate the computational dynamics during population bursting as well as the different roles that might be played by various nodes during these bursts. This is due to its ability to analyse information processing *locally in time* (*Lizier, 2013*; *Lizier et al., 2008*; *Lizier, 2014*; *Wibral et al., 2014b*), as well as directionally between information sources and targets via the asymmetry of TE. This allows us to isolate the information

processing taking place during population bursting activity. We can then determine the information processing roles undertaken by the different nodes and examine how this relates to their average position in the burst propagation.

We analyse the information flowing between nodes during population bursts by estimating the *burst-local* TE between nodes in each recording (i.e. averaging the TE rates only during bursting periods, using probability distribution functions estimated over the whole recordings; see section 'Estimation of burst-local TE'). We also measure the mean position of each node within bursts (with earlier bursting nodes having a lower numerical position; see section 'Analysis of population bursts'). Note that, although there is variability of the burst position across bursts, certain nodes have much lower or higher mean burst positions, indicating a strong tendency to burst earlier or later within the propagation.

*Figure 5a and b* show plots of the mean burst position plotted against the total inward (*Figure 5a*) and outward (*Figure 5b*) burst-local TE of each node. *Appendix 3—figure 7* shows these plots for the additional cultures. Plots are only shown for days where there were at least 10 statistically significant burst-local TE values. The Spearman correlation ($\rho$) between these variables is also displayed on the plots.

We see from *Figure 5* that, particularly for the final recording days, in most cases there is a positive correlation between the mean burst position of the node and the total inward burst-local TE. In some cases, this correlation is particularly strong. For instance, on the 24th DIV of culture 1-3, we observe a Spearman correlation of $\rho = 0.84$. In other words, we observe that nodes which tend to burst later have higher incoming information flows. *Table 4* summarises the proportions of recordings (including the additional cultures) which had significant positive Spearman correlations between the mean burst position of the node and the total inward burst-local TE. By focussing on recordings that have reached a state of established information dynamics, by either selecting all final recordings or all that were performed post day 15, we see that in all cases a clear majority of cases had a statistically significant positive Spearman correlation. Moreover, the probability of this number of correlations arising by chance is found to be very low. These relationships suggest that there is a tendency for the late bursters to occupy the specialised computational role of information receivers.

Conversely, as shown in *Figure 5*, there is a tendency for the mean burst position of the nodes to be negatively correlated with the outward burst-local TE. Again, this correlation is particularly strong in many cases. For example, on the 24th DIV of culture 1-3, we observe a Spearman correlation of $\rho = -0.80$. In other words, we observe that nodes which tend to burst earlier have higher outward information flows. *Table 4* summarises the proportions of recordings (including the additional cultures) which had significant negative Spearman correlations between the mean burst position of the node and the total outward burst-local TE. We see that a clear majority of either all final recording days or all recordings performed post day 15 had a statistically significant negative Spearman correlation. Moreover, the probability of this number of correlations arising by chance is found to be very low. These relationships suggest that there is a tendency for the early bursters to occupy the specialised computational role of information receivers.

*Figure 5* plots the total incoming burst-local TE on each node against the total outgoing burst-local TE, with points coloured according to the node's mean burst position (*Appendix 3—figure 8* shows these plots for the additional cultures). We see a very clear pattern in these plots, which is remarkably clear on some later recording days: nodes which often fire at the beginning of the burst progression have high outgoing information flows with lower incoming flows, whereas those which tend to sit at the end of the progression have high incoming flows with lower outgoing flows. By contrast, those nodes which, on average, occupy the middle of the burst progression have a balance between outgoing and incoming information transfer. These nodes within the middle of the burst propagation are, therefore, occupying the suggested role of mediators of information flow.

## Early lock-in of specialised computational roles

Given that we have seen in section 'Information flows quantify computational role of burst position' that nodes tend to occupy specialised computational roles based on their average position in the burst propagation and that we have seen in section 'Early lock-in of information flows' that information processing properties can lock-in early in development, it is worth asking whether the specialised

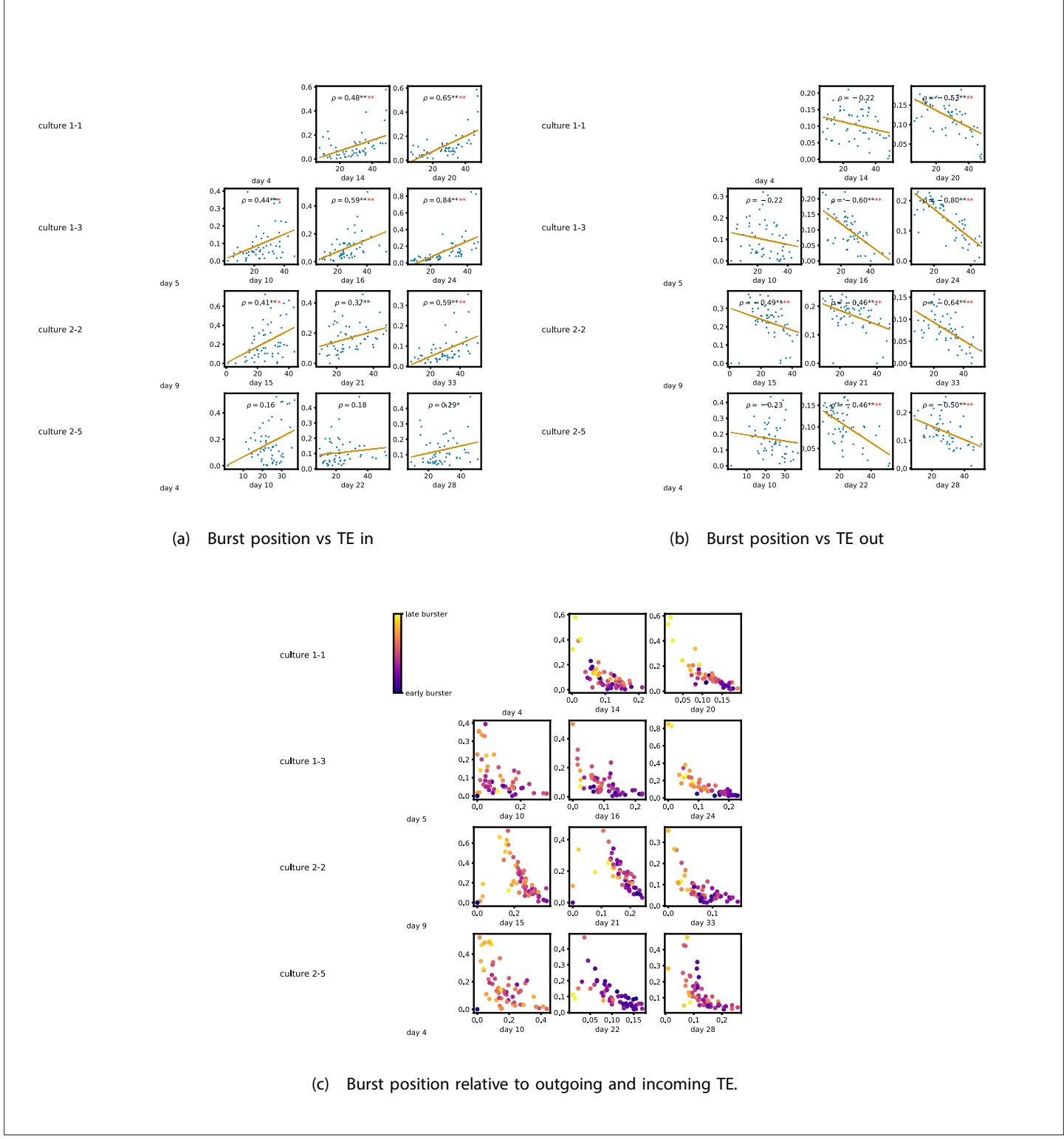

**Figure 5.** The relationship between the amount of incoming and outgoing local (in burst) transfer entropy (TE) on a given node and its average burst position. (**a**) and (**b**) show the burst position of each node on the $x$ axis of each plot, plotted against either the total incoming (**a**) or outgoing (**b**) TE on the node. The Spearman correlation ($\rho$) between the mean burst position and the incoming or outgoing TE values is displayed in each plot. Values of $\rho$ significant at the 0.05 level are designated with an asterisk and those significant at the 0.01 level are designated with a double asterisk. Red asterisks are used to denote significance after performing a Bonferroni correction for multiple comparisons. (**e**) plots the outgoing TE on the $x$ axis and the incoming TE on the $y$ axis with the points coloured according to the mean burst position of the node: late bursters are coloured yellow and early bursters are purple.

**Table 4.** Summary of significance values for the results relating to computational roles. Each table cell shows the number of relationships that were found to be significant at the $p<0.05$ level, out of the total number of relationships tested. A hypothesis test is conducted against the null hypothesis that the original p-values that produced these results were uniformly distributed between 0 and 1 (giving a 0.05 chance of a significant result). ** indicates that the probability of the observed number of significant results has probability less than 0.001 under this null hypothesis, with a Bonferroni correction for multiple comparisons. The first row summarises the significance values for the relationships between inward burst-local TE and burst position, as shown in *Figure 5* and *Appendix 3—figure 7*. The second row summarises the significance values for the relationships between outward burst-local TE and burst position, as shown in *Figure 5* and *Appendix 3—figure 7*.

|  | All cultures, final day | All cultures post day 15 | Bursting, final day | Bursting, post day 15 |
|---|---|---|---|---|
| In | 7/11** | 10/15** | 7/10** | 10/14** |
| Out | 7/11** | 11/15** | 7/10** | 11/14** |

computational roles that nodes occupy during population bursts lock-in during the earlier stages of neuronal development.

In order to investigate this question, we quantified the computational role occupied by a node by measuring the proportion of its total incoming and outgoing burst-local TE that was made up by its outgoing burst-local TE. Scatters of these proportions between earlier and later DIV are plotted in *Figure 6* for the main cultures (the additional cultures are shown in *Appendix 3—figure 9*). They also display the Spearman rank-order correlations ($\rho$) between the ratios on different days. We see that, in many cases, there are strong, significant, correlations in this ratio between earlier and later DIV. *Table 3* summarises the proportions of pairs of recordings (including the additional cultures) which had significant positive Spearman correlations between this ratio on each node across days. We see that, whether we focus on just the last recordings of each culture or also include those after day 15, a clear majority of pairs of recordings exhibit such correlations and that such a majority would be very unlikely to arise by chance. *Figure 6* visualises all these Spearman correlations between the early and late day pairs.

These results suggest that, if a node is an information transmitter during population bursts at the point at which the information flows are established, it has a tendency to maintain this specialised role later in development. Similarly, being an information receiver earlier in development increases the probability that the node will occupy this same role later.

## Information flows in an STDP model of development

In order to investigate the generality of the phenomena revealed in this article, we reimplemented a model network (*Khoshkhou and Montakhab, 2019*) of Izhikevich neurons (*Izhikevich, 2003*) developing according to an STDP (*Caporale and Dan, 2008*) update rule as described in section 'Network of Izhikevich neurons'. For the low value of the synaptic time constant which we used (see section 'Network of Izhikevich neurons'), these networks developed from a state where each neuron underwent independent tonic spiking at a regular firing rate to one in which the dynamics were dominated by periodic population bursts (*Zeraati et al., 2021*; *Khoshkhou and Montakhab, 2019*). It is worth noting that these population bursts are significantly more regular than those in the biological data used in this article. Small modifications were made to the original model in order that the development occurred over a greater length of time. The greater length of development allowed us to extract time windows which were short relative to the timescale of development (resulting in the dynamics being approximately stationary in these windows) yet still long enough to sample enough spikes for reliable TE rate estimation. The windows which we used resulted in a median of 5170 spikes per neuron per window compared with a median of 17,399 spikes per electrode in the biological data. See section 'Network of Izhikevich neurons' for more details on the modifications made. A single simulation was run. The dynamics of the model are very consistent across independent runs. Three windows were extracted, extending between the simulation timepoints of 200 and 250 seconds, 400 and 450 seconds, and 500 and 550 seconds. These time windows were labelled 'early', 'mid', and

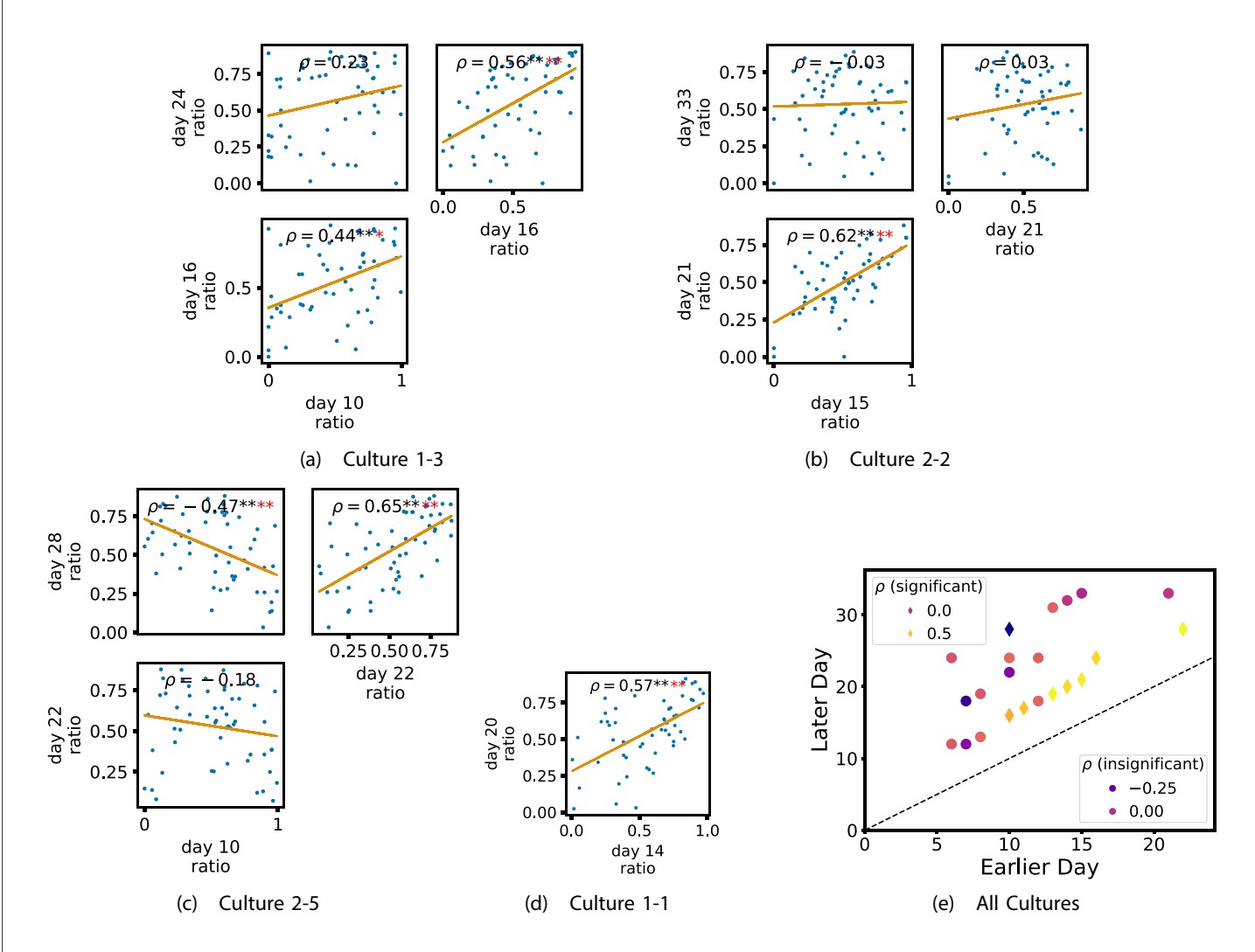

**Figure 6.** Plots investigating the relationship between the ratio of outward to total burst-local information flow from a given node over different days of development. (**a–d**) show scatter plots between all pairs of days for each culture (excluding days with less than 10 significant burst-local transfer entropy [TE] values). Specifically, in each scatter plot, the $x$ value of a given point is the ratio of total outgoing burst-local TE on the associated node to the total burst-local TE on the same node on one day and the $y$ value of that same point is this same ratio on the same node but on a different day. The days in question are shown on the bottom and sides of the grids of scatter plots. The orange line shows the ordinary least-squares regression. The Spearman correlation ($\rho$) between the TE values on the two days is displayed in each plot. Values of $\rho$ significant at the 0.05 level are designated with an asterisk and those significant at the 0.01 level are designated with a double asterisk. Red asterisks are used to denote significance after performing a Bonferroni correction for multiple comparisons. (**e**) shows all recording day pairs for all cultures (where the pairs are always from the same culture) and the associated Spearman correlation between the outward TE of the nodes across this pair of recording days. Diamonds indicate significance at p<0.05, with Bonferroni correction.

'late', respectively. The early window was chosen such that it had a nonzero number of significant TE values, but such that this number was of the same (order of magnitude in) proportion as observed in the first recording days of the cell cultures (refer to *Table 2*). The mid period was set at the point where population bursting begun to emerge, and the late period was set at the point where all neurons were bursting approximately synchronously in a pronounced manner.

TE values between all pairs of model neurons were estimated, as described in section 'Transfer entropy estimation'. These estimates were then subjected to the same statistical analysis as the cell culture data, the results of which are presented in the preceding subsections of this section. The plots of this analysis are displayed in *Figures 7 and 8*.

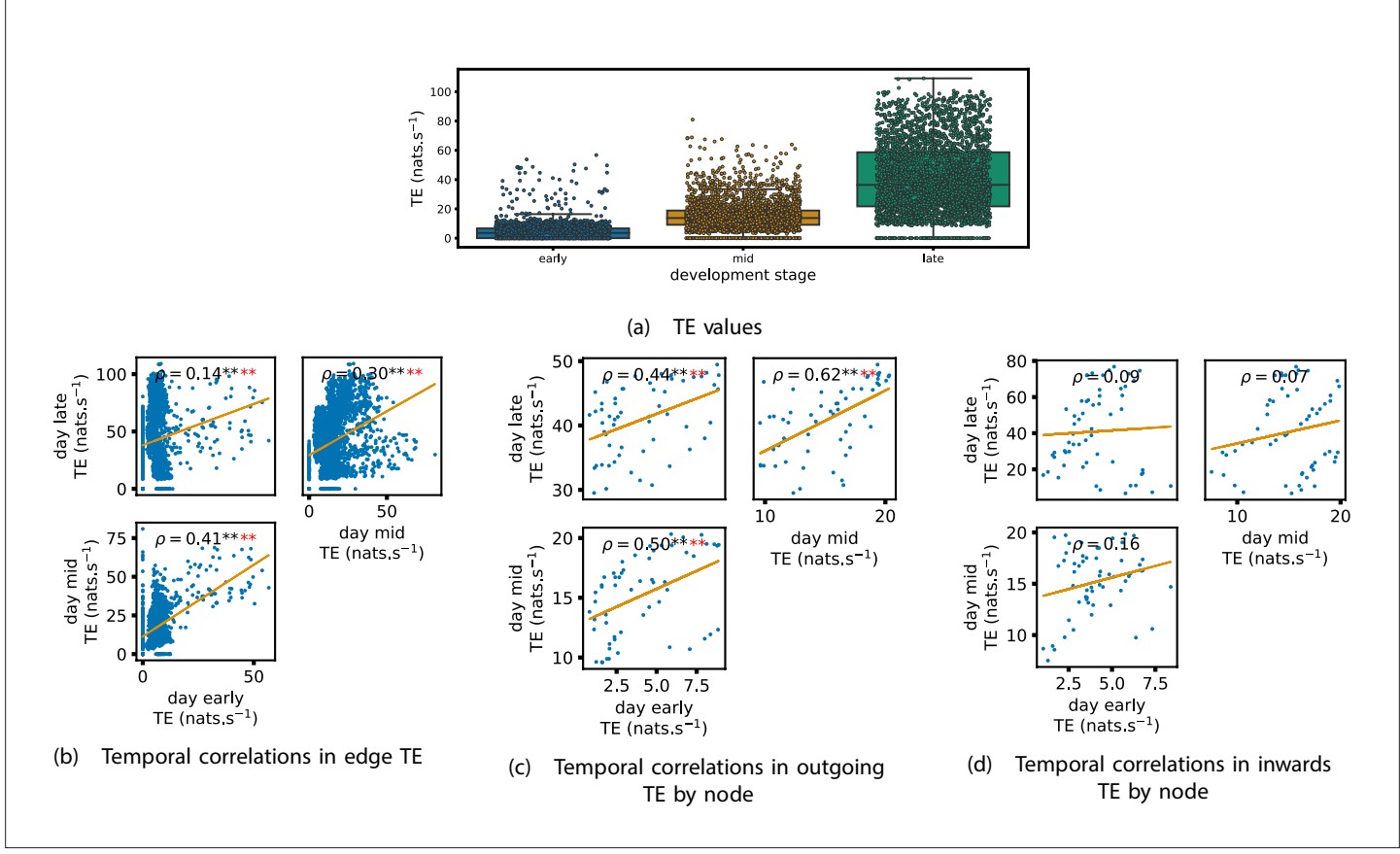

**Figure 7.** Equivalent plots to those shown in *Figures 1, 3 and 4* and *Appendix 2—figure 1*, but for the simulated spiking network developing under spike-timing-dependent plasticity (STDP). (**a**) shows scatters of the transfer entropy (TE) values overlaid on box plots. The box plots show the quartiles and the median (values greater than 10 SDs from the mean have been removed from both the box and scatter plots as outliers). It corresponds to *Figure 1*. (**b**–**d**) show scatter plots investigating the relationship between TE values (or derived summary statistics) over different stages of development. Specifically, in each scatter plot, the *x* value of a given point is a TE value or derived statistic at an earlier simulation stage and the *y* value of that same point is a TE value (or derived statistic) on the corresponding edge or node, but later in the simulation. The orange line shows the ordinary least-squares regression. The Spearman correlation ($\rho$) between the TE values on the two days is displayed in each plot. Values of $\rho$ significant at the 0.05 level are designated with an asterisk and those significant at the 0.01 level are designated with a double asterisk. Red asterisks are used to denote significance after performing a Bonferroni correction for multiple comparisons. (**b**) corresponds to the scatter plots in *Figure 3*, (**c**) corresponds to the scatter plots in *Figure 4*, and (**d**) corresponds to the scatter plots in *Appendix 2—figure 1*.

Scatters and box plots of the TE values estimated in each developmental window are shown in *Figure 7*. We observe a large, monotonic, increase in these values over development. This mirrors the finding in cell cultures, as described in section 'The dramatic increase in the flow of information during development'.

We also observe a similar lock-in phenomenon of information processing as was found in the cell cultures (described in section 'Early lock-in of information flows'). *Figure 7a–d* show the correlation in information flow between different stages of development. Specifically, *Figure 7a* plots the correlation in TE values between each ordered pair of neurons between early and later windows. *Figure 7d* plots this same correlation, but for the total incoming TE on each neuron, and *Figure 7c* does this for the total outgoing TE. In all six of the plots for the relationships between the TE on each edge and for the total outgoing TE, we observe a substantial statistically significant positive correlation between values on earlier and later days (significant at the p<0.01 level, with Bonferroni correction). We observe smaller positive correlations in these values for the total incoming TE on each node, although these correlations are not significant, which aligns somewhat with the weaker effect observed for incoming TE in the cultures. As with the cell cultures, some of the observed correlations are particularly strong, such as the Spearman correlation of $\rho = 0.62$ between the total outgoing TE on each node in the mid window and this same value in the late window. This implies that the spatial structure of the

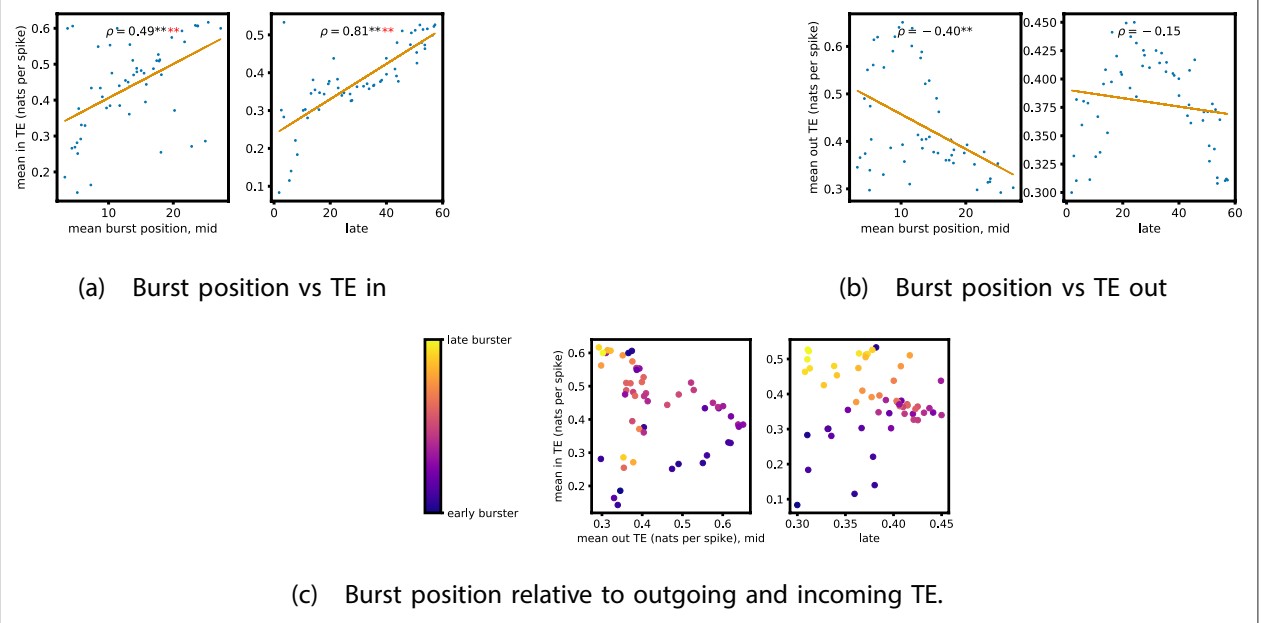

(a) Burst position vs TE in

(b) Burst position vs TE out

(c) Burst position relative to outgoing and incoming TE.

**Figure 8.** Equivalent plots to those shown in *Figure 5*, but for the simulated spiking network developing under spike-timing-dependent plasticity (STDP). Plots show the relationship between the amount of incoming and outgoing local (in burst) transfer entropy (TE) on a given node and its average burst position. (**a**) and (**b**) show the burst position of each node on the $x$ axis of each plot, plotted against either (**a**) the total incoming or (**b**) outgoing TE on the node. The Spearman correlation ($\rho$) between the mean burst position and the incoming or outgoing TE values is displayed in each plot. Values of $\rho$ significant at the 0.05 level are designated with an asterisk and those significant at the 0.01 level are designated with a double asterisk. Red asterisks are used to denote significance after performing a Bonferroni correction for multiple comparisons. (**c**) plots the outgoing TE on the $x$ axis and the incoming TE on the $y$ axis with the points coloured according to the mean burst position of the node: late bursters are coloured yellow and early bursters are purple.

information flow has a tendency to be determined in the earlier stages of development, after which they are locked-in – in a similar fashion to what was observed in the biological experiments in earlier sections.

We also performed the same analysis on computational roles as presented in section 'Information flows quantify computational role of burst position'. This analysis, the results of which are presented in *Figure 8*, only looked at the mid and late windows. The early window was ignored due to its lack of bursting activity. In the mid recording window, we observe a strong relationship between the mean burst position of the neuron and its computational role. *Figure 8* shows that there is a significant (at the $p<0.01$ level) positive correlation between the mean burst position of a neuron and its total incoming burst-local TE (see section 'Estimation of burst-local TE' for more details on the burst-local TE). There is also a negative correlation between the mean burst position and the total outgoing burst-local TE, as shown in *Figure 8*. However, this relationship is not significant after Bonferroni correction. These same figures also display these relationships for the late window. Here, we observe the same directions of relationships. This relationship is incredibly strong between incoming TE and burst position ($\rho = 0.81$). The relationship between the outgoing TE and burst position is still negative, although it is not as strong as in the mid window and is no longer significant. Inspection of *Figure 8* reveals that there is still a very clear negative relationship between burst position and outgoing TE after a burst position of about 20. Indeed, if we condition on the burst position being greater than 20, then we find a Spearman correlation of $\rho = 0.80$, which is significant at the $p<0.01$ with Bonferroni correction. Inspection of the spike rasters of these simulations suggests that the anomalous results for the earliest bursters may be due to their spiking a very substantial distance ahead of the rest of the population in these simulations. These differing burst dynamics mean that the earliest bursters are then less able to reduce the uncertainty in the spike times of the majority of the neurons which begin spiking significantly later.

This implies that, midway through development, we are observing the same specialisation into computational roles based on burst position as was observed in the cell cultures: early bursters display

a tendency to be information transmitters, late bursters operate as receivers, and middle bursters exhibit a balance of the two. Later on in development, we do, however, observe a slight departure from the roles we observed in the cell cultures. The computational roles are shifted further down the burst propagation and the earliest bursters here are less strongly driving the rest of the population.

It is worth noting that the estimated TE values in the model are substantially higher than in the biological dataset. The median estimated TE in the late window of the model was around $40\,\mathrm{nats.s^{-1}}$ (*Figure 7*). Conversely, it was less than $0.2\,\mathrm{nats.s^{-1}}$ for every last recording day of the cell cultures (*Figure 1*). This is due to the much higher spike rate of the model implying that the dynamics are operating on different timescales. Indeed, if we compare the magnitude of the burst-local TE – which is measured in nats per spike (see section 'Estimation of burst-local TE') – between the model and the biological data (*Figures 8 and 5*, respectively), we find values of similar magnitude.

In summary, in a network model of Izhikevich neurons developing according to STDP towards a state of population bursts, we observe a similar developmental information processing phenomena as in the cell cultures. Namely, the amount of information flowing across the network increases dramatically, the spatial structure of this flow locks in early, and the neurons take on specialised computational roles based on their burst position.

## Discussion

Biological neural networks are imbued with an incredible capacity for computation, which is deployed in a flexible manner in order to achieve required tasks. Despite the importance of this capacity to the function of organisms, how it emerges during development has remained largely a mystery. Information dynamics (*Lizier, 2013*; *Lizier et al., 2014*; *Lizier et al., 2008*; *Lizier et al., 2010*; *Lizier et al., 2012*) provides a framework for studying such computational capacity by measuring the degree to which the fundamental information processing operations of information storage, transfer, and modification occur within an observed system.

Previous work on the information flow component of computational capacity in neural cell cultures (*Nigam et al., 2016*; *Shimono and Beggs, 2015*; *Matsuda et al., 2013*; *Timme et al., 2014*; *Kajiwara et al., 2021*; *Timme et al., 2016*; *Wibral et al., 2017*) has focussed on the static structure of information flow networks at single points in time. This has mostly taken the form of elucidating properties of the functional networks implied by the information flows. However, such work leaves open questions concerning how these structures are formed. We address this gap here.

An initial goal in addressing how computational capacity emerges was to determine when the information flow component arrived. It is plausible that this capacity could have been present shortly after plating or that it could have arrived suddenly at a later point in maturation. What we see, however, is that the capacity for information transmission is either not present or only minimally present in the early DIV. This can be seen by looking at the very low mean TE values in the first column of *Table 1*. However, over the course of development we see that the TE values increase progressively, reaching values orders of magnitude larger. This implies that information transmission is a capacity which is developed enormously during neuronal development and that its gain is spread consistently throughout the observed period.

The information processing operations of a system tend to be distributed over it in a heterogeneous fashion. For example, it has been found in models of whole-brain networks (*Li et al., 2019*; *Marinazzo et al., 2012*; *Marinazzo et al., 2014b*), abstract network models (*Ceguerra et al., 2011*; *Novelli et al., 2020*; *Goodman and Porfiri, 2020*), and even energy networks (*Lizier et al., 2009*) that nodes with high indegrees tend to also have high outgoing information flows. Section 'The emergence of functional information flow networks' examined the emergent information flow networks, formed by connecting nodes with a statistically significant TE value between them. In accordance with this previous work – and indeed the large variation in shared, unique and synergistic information flow components observed on the same dataset (albeit with the discrete-time estimator) (*Wibral et al., 2017*) – these networks exhibited a high degree of heterogeneity. Notably, as shown in *Figure 2a*, they have prominent hubs of inward flow (sinks) along with less pronounced hubs of outgoing flow (sources). Moreover, along with heterogeneity within individual networks, large structural differences are easily observed between the different networks shown in *Figure 2a*.

Keeping with our goal of uncovering how features of mature information flow networks self-organise, we examined how this heterogeneity at both the intra-network and inter-network levels

emerged. It was found in section 'Early lock-in of information flows' that the key features of the information flow structure are locked-in early in development, around the point at which the information flows emerge. This effect was identified for the outgoing TE from each node, for example, where we found strong correlations over the different days of development. It is worth further noting that this lock-in phenomenon occurs remarkably early in development. Specifically, in very many cases, we observe strong correlations between quantities estimated on the first recording days with nonzero TE and the same quantities estimated on later days. This early lock-in provides us with a mechanism for how the high heterogeneity exhibited in the inflow and outflow hubs emerges. Small differences between networks on early DIV will be magnified on subsequent days. This leads to the high levels of inter-network heterogeneity that we observe. A similar phenomenon has been observed with STDP, which can lead to symmetry breaking in network structure (*Gilson et al., 2009*; *Kunkel et al., 2011*), whereby small fluctuations in early development can set the trajectory of the synaptic weights on a specific path with a strong history dependence. In order to confirm a hypothesis that this observed lock-in of information flows could be induced by STDP, in section 'Information flows in an STDP model of development' we studied the information dynamics of a model network of Izhikevich neurons developing according to an STDP (*Caporale and Dan, 2008*) update rule from a state of independent tonic firing to population bursting. The lock-in of key features of the information flow structure was evident over the period where the network developed from independent firing to approximately synchronous bursting (i.e., bursts occurring at approximately the same point in time). This indicates a plausible mechanism for our observations and suggests a broader generality of these phenomena.

An interesting difference between the results for the model and the biological data is that the lock-in effect was stronger for outward TE as well as the TE on the edges than in the biological data. The reasons for this difference require further investigation; however, it might be due to the multi-unit nature of the biological data. A possible direction of future work is to modify the model such that the activity on neurons is sub-sampled and aggregated in order to model the effect of placing electrodes in a culture. However, this will detract from the simplicity of the model in its current form.

It has been hypothesised that different neural units take on specialised computational roles (*Schroeter et al., 2015*; *Frost and Goebel, 2012*; *Cohen and D'Esposito, 2016*). In section 'Information flows quantify computational role of burst position', we investigated the information flows occurring during the critical bursting periods of the cultures' dynamics. Specifically, we studied the burst-local TE in order to measure the information being transferred between nodes during these periods. The plots shown in *Figure 5* show a clear tendency for the nodes to take on specialised computational roles as development progresses. Moreover, these computational roles were tightly coupled to the position the node tended to burst within in the burst propagation. Nodes that tended to initiate the bursts had a tendency to have high outgoing information transfer combined with low incoming information flow, implying their role as information transmitters. The opposite relationship is observed for typically late bursters, indicating their role as information receivers. By contrast, nodes that usually burst during the middle of the progression have a balance between outward and inward flows. This indicates that they are the crucial links between the transmitters and receivers of information. Neurons bursting in the middle of the burst progression of dissociated cell cultures have received special attention in past work using undirected measures, where it was conjectured that they act as the 'brokers of neuronal communication' (*Schroeter et al., 2015*). In this work, we have provided novel supporting evidence for this conjecture by specifically identifying the *directed* information flows into and out of these nodes. Moreover, in section 'Information flows in an STDP model of development', we observed that this same specialisation of neurons into computational roles based on average burst position occurred in a model network of Izhikevich neurons which had developed via an STDP learning rule to a state of population bursting. This suggests that this phenomenon might exist more generally than the specific cell cultures studied. It is also worth noting that some of these relationships, notably those shown in *Figure 7b and d*, are much stronger than what was observed in the cell culture. It is likely that this is due to the fact that in the model we estimated TE between individual model neurons, whereas in the cultures we estimated TE between the MUA on each electrode. A possible direction for future work will be to study how the estimated information flow changes when we aggregate the spikes from multiple model neurons into simulated MUA.

It is worth reflecting on the fact that the observed correlations between burst-local information transfer and average burst position will not occur in all neuronal populations. For instance, in

populations with strictly periodic bursts, each node's behaviour will be well explained by its own history, resulting in very low burst-local TE's, regardless of burst position. Furthermore, neuronal populations develop bursty dynamics to different extents and such quantities (let alone their correlation) are simply less meaningful in the absence of burstiness (e.g. for the final day of culture 1-5 as stated in Figure 3A of *Wagenaar et al., 2006b*).

Returning once more to our focus on investigating the emergence of information flows, we have demonstrated, in section 'Early lock-in of specialised computational roles', that these specialist computational roles have a tendency to lock-in early. There we looked at the ratio of outgoing burst-local TE to the total burst-local TE on each node. It was found that there is a strong tendency for this ratio to be correlated between early and late days of development. This suggests that the computational role that a node performs during population bursts is determined to a large degree early in development.

Insights into development aside, a fundamental technical difference between the work presented here and previous studies of TE in neural cultures is that here we use a recently developed continuous-time estimator of TE (*Shorten et al., 2021*). This estimator was demonstrated to have far higher accuracy in estimating information flows than the traditional discrete-time estimator. The principal challenge which is faced when using the discrete-time estimator is that the curse of dimensionality limits the number of previous time bins that can be used to estimate the history-dependent spike rates. All applications of this estimator to spiking data from cell cultures of which the authors are aware (*Nigam et al., 2016*; *Shimono and Beggs, 2015*; *Matsuda et al., 2013*; *Timme et al., 2014*; *Kajiwara et al., 2021*; *Timme et al., 2016*) made use of only a single previous bin in the estimation of these rates. This makes it impossible to simultaneously achieve high time precision and capture the dependence of the spike rate on spikes occurring further back in time. Conversely, by operating on the interspike intervals, the continuous-time estimator can capture the dependence of the spike rate on events occurring relatively far back in time, while maintaining the time precision of the raw data. Looking at a specific representative example, our target history embeddings made use of the previous four interspike intervals (section 'Selection of embedding lengths'). For the recording on day 24 of culture 1-3, the mean interspike interval was 0.71 s. This implies that the target history embeddings on average extended over a period of 2.84 s. The raw data was collected with a sampling rate of 25 kHz (*Wagenaar et al., 2006b*). In order to lose no time precision, the discrete-time estimator would thus have to use bins of 40 s, and then in order to extend over 2.84 s, the target history embeddings would therefore need to consist of around 70,000 bins which is empirically not possible to sample well.

It is worth noting that, as we were performing a longitudinal analysis where each studied recording was separated by days or weeks, we did not perform spike sorting as we would have been unable to match the different units on an electrode across different recordings. We would then not have been able to compare the TE values on a given unit over the course of development. Instead, we analysed the spikes on each electrode without sorting. As such, this work studies MUA (*Schroeter et al., 2015*). Spike sorting applied to data collected from a near-identical recording setup found an average of four neurons per electrode (*Wagenaar et al., 2006a*). This situates this work at a spatial scale slightly larger than spike-sorted neural data, but still orders of magnitude finer than fMRI, EEG, or MEG (*Bassett and Sporns, 2017*).

Functional and effective networks arising from the publicly available dataset used in this article (*Wagenaar et al., 2006b*) have been studied by other authors. For instance, it was shown that the functional networks were small world (*Downes et al., 2012*) and that more connected nodes exhibited stronger signatures of nonlinearity (*Minati et al., 2019*). Work has also been conducted analysing the burst propagation of these cultures, finding that there are 'leader' nodes which consistently burst before the rest of the population (*Eckmann et al., 2008*). These are the information transmitters that we have observed in this work here.

An exciting direction for future work will be to move beyond directed functional networks to examine the information flow provided by higher-order multivariate TEs in an effective network structure (*Novelli and Lizier, 2021*; *Novelli et al., 2019*). The networks inferred by such higher-order TEs are able to better reflect the networks' underlying structural features (*Novelli and Lizier, 2021*). As was the case with bivariate TEs prior to this work, there is an absence of work investigating how the networks of multivariate information flow emerge during neural development. Moreover, moving to higher-order measures will allow us to more fully characterise the multifaceted specialised computational roles undertaken by neurons.

**Table 5.** File numbers used for each culture on each day.
These correspond to the file numbering used in the freely available dataset used in this study,
provided by *Wagenaar et al., 2006b*; *Network, 2021*.

| Culture 1-1 | Day 4 | Day 14 | Day 20 | |
|---|---|---|---|---|
| | 2 | 2 | 2 | |
| Culture 1-2 | Day 6 | Day 11 | Day 17 | |
| | 2 | 2 | 2 | |
| Culture 1-3 | Day 5 | Day 10 | Day 16 | Day 24 |
| | 2 | 2 | 2 | 2 |
| Culture 1-4 | Day 8 | Day 13 | Day 19 | |
| | 2 | 2 | 2 | |
| Culture 1-5 | Day 7 | Day 12 | Day 18 | |
| | 2 | 2 | 2 | |
| Culture 2-1 | Day 14 | Day 32 | | |
| | 2 | 2 | | |
| Culture 2-2 | Day 9 | Day 15 | Day 21 | Day 33 |
| | 2 | 2 | 2 | 2 |
| Culture 2-3 | Day 6 | Day 12 | Day 24 | |
| | 2 | 2 | 2 | |
| Culture 2-4 | Day 3 | Day 5 | Day 11 | |
| | 1 | 1 | 1 | |
| Culture 2-5 | Day 4 | Day 10 | Day 22 | Day 28 |
| | 1 | 1 | 2 | 1 |
| Culture 2-6 | Day 7 | Day 13 | Day 31 | |
| | 1 | 1 | 1 | |

# Materials and methods
## Cell culture data

The spike train recordings used in this study were collected by *Wagenaar et al., 2006b* and are freely available online (*Network, 2021*). The details of the methodology used in these recordings can be found in the original publication (*Wagenaar et al., 2006b*). A short summary of their methodology follows.

Dissociated cultures of rat cortical neurons had their activity recorded. This was achieved by plating 8 × 8 MEAs, operating at a sampling frequency of 25 kHz with neurons obtained from the cortices of rat embryos. The spacing between the electrodes was 200 m centre-to-centre. The MEAs did not have electrodes on their corners and one electrode was used as ground, resulting in recordings from 59 electrodes. In all recordings, electrodes with less than 100 spikes were removed from the analysis. This resulted in electrodes 37 and 43 (see *Figure 2* for the position of these electrodes) being removed from every recording as no spikes were recorded on them. The spatial layout of the electrodes is available from the website associated with the dataset (*Network, 2021*), allowing us to overlay the functional networks onto this spatial layout as is done in *Figure 2a*.

30 min recordings were conducted on most days, starting from 3 to 4 DIV. The end point of recording varied between 25 and 39 DIV. Longer overnight recordings were also conducted on some cultures at sparser intervals. As the accurate estimation of information-theoretic quantities requires substantial amounts of data (*Shorten et al., 2021*; *Kraskov et al., 2004*), in this work we make use

of these longer overnight recordings. These recordings were split into multiple files. The specific files used, along with the names of the cultures and days of the recordings, are listed in *Table 5*.

The original study plated the electrodes with varying densities of cortical cells. However, overnight recordings were only performed on the 'dense' cultures, plated with a density of $2500\,\mathrm{cells/L}$.

The original study performed threshold-based spike detection by determining that a spike was present in the case of an upward or downward excursion beyond 4.5 times the estimated RMS noise of the recorded potential on a given electrode. The analysis presented in this article makes use of these detected spike times. No spike sorting was performed, and, as such, we are studying MUA (*Schroeter et al., 2015*).

## Network of Izhikevich neurons

The model spiking network used to generate the data analysed in section 'Information flows in an STDP model of development' is identical to that presented in *Khoshkhou and Montakhab, 2019*, with a few minor alterations. This model consists of Izhikevich neurons (*Izhikevich, 2003*) developing according to an STDP (*Caporale and Dan, 2008*) update rule. At the beginning of the simulation, each neuron performs independent tonic spiking; however, the network develops towards population bursts.

The specific model settings used were based on those used to produce Figure 5A in *Khoshkhou and Montakhab, 2019*. That is, the proportion of inhibitory neurons ($\alpha$) and the synapse time delay ($\tau_{ij}$) were both set to 0. The first change made was to use 59 neurons, as opposed to the 500 used in *Khoshkhou and Montakhab, 2019*, in order to correspond to the number of electrodes used in the cell culture recordings. The maximum connection strength ($g_{max}$) was also increased from 0.6 to 10 in order to compensate for this reduction in the network size.

The only remaining change was made in order to slow the rate of development of the population. The reasoning behind this was to allow for the extraction of windows which were much shorter than the timescale of development, resulting in the dynamics within these windows being approximately stationary (and including enough samples for estimation of the TE rates). Specifically, this change was to greatly reduce the values of the maximum synaptic potentiation and depression ($A_+$ and $A_-$). These values were reduced from $5 \times 10^{-2}$ to $4 \times 10^{-4}$.

## Data preprocessing

As the data was sampled at 25 kHz, uniform noise distributed between –20 s and 20 s was added to each spike time. This is to prevent the TE estimator from exploiting the fact that, in the raw data, interspike intervals are always an integer multiple of 40 s.

## Transfer entropy estimation

The (bivariate) TE (*Schreiber, 2000*; *Bossomaier et al., 2016*) was estimated between each pair of electrodes in each of the recordings listed in *Table 5*. TE is the mutual information between the past state of a source process and the present state of a target process, conditioned on the past state of the target. More specifically (in discrete time), the TE rate is

$$\dot{\mathbf{T}}_{Y \to X} = \frac{1}{\Delta t} I\left(X_t \,;\, \mathbf{Y}_{<t} \,\middle|\, \mathbf{X}_{<t}\right) = \frac{1}{\tau} \sum_{t=1}^{N_T} \ln \frac{p\left(x_t \,\middle|\, \mathbf{x}_{<t}, \mathbf{y}_{<t}\right)}{p\left(x_t \,\middle|\, \mathbf{x}_{<t}\right)}. \tag{1}$$

The TE above is being measured from a source $Y$ to a target $X$, $I(\cdot\,;\,\cdot|\cdot)$ is the conditional mutual information (*MacKay and Kay, 2003*), $x_t$ is the current state of the target, $\mathbf{x}_{<t}$ is the history of the target, $\mathbf{y}_{<t}$ is the history of the source, $\Delta t$ is the bin width (in time units), $\tau$ is the length of the processes, and $N_T = \tau/\Delta t$ is the number of time samples (bins). The histories $\mathbf{x}_{<t}$ and $\mathbf{y}_{<t}$ are usually captured via embedding vectors, for example, $\mathbf{x}_{<t} = \mathbf{x}_{t-m:t-1} = \{x_{t-m}, x_{t-m+1}, \ldots, x_{t-1}\}$.

### Previous application of the discrete-time estimator

Previous applications of TE to spiking data from neural cell cultures (*Nigam et al., 2016*; *Shimono and Beggs, 2015*; *Matsuda et al., 2013*; *Timme et al., 2014*; *Kajiwara et al., 2021*; *Timme et al., 2016*; *Wibral et al., 2017*) made use of this discrete-time formulation of TE. This work was primarily focussed on the directed functional networks implied by the estimated TE values between pairs of

nodes which has revealed interesting features of the information flow structure. *Shimono and Beggs, 2015* found that these networks exhibited a highly non-random structure and contained a long-tailed degree distribution. This work was expanded by *Nigam et al., 2016*, where it was found that the functional networks contained a rich-club topology. Conversely, *Timme et al., 2014* found that the hubs of these networks were localised to certain timescales. Other work (*Timme et al., 2016*; *Wibral et al., 2017*) has instead focussed on how the components of information flows in cell cultures can be decomposed into unique, redundant, and synergistic components.

## Continuous-time estimation

It has, relatively recently, been shown that, for event-based data such as spike trains, in the limit of small bin size, the TE is given by the following expression (*Spinney and Lizier, 2018*):

$$\dot{\mathbf{T}}_{Y \to X} = \lim_{\tau \to \infty} \frac{1}{\tau} \sum_{i=1}^{N_X} \ln \frac{\lambda_{x|\mathbf{x}_{<t},\mathbf{y}_{<t}} \left[ \mathbf{x}_{<x_i}, \mathbf{y}_{<x_i} \right]}{\lambda_{x|\mathbf{x}_{<t}} \left[ \mathbf{x}_{<x_i} \right]}. \tag{2}$$

Here, $\lambda_{x|\mathbf{x}_{<t},\mathbf{y}_{<t}} \left[ \mathbf{x}_{<x_i}, \mathbf{y}_{<x_i} \right]$ is the instantaneous firing rate of the target conditioned on the histories of the target $\mathbf{x}_{<x_i}$ and source $\mathbf{y}_{<x_i}$ at the time points $x_i$ of the spike events in the target process. $\lambda_{x|\mathbf{x}_{<t}} \left[ \mathbf{x}_{<x_i} \right]$ is the instantaneous firing rate of the target conditioned on its history alone, ignoring the history of the source. It is important to note that the sum is being taken over the $N_X$ spikes of the target, thereby evaluating log ratios of the expected spike rates of the target given source and target histories versus target histories alone, *when* the target does spike. As this expression allows us to ignore the 'empty space' between events, it presented clear potential for allowing for more efficient estimation of TE on spike trains.

This potential was recently realised in a new continuous-time estimator of TE presented in *Shorten et al., 2021* (and utilised in *Mijatovic et al., 2021*), and all TE estimates in this article were performed using this new estimator. In *Shorten et al., 2021* it is demonstrated that this continuous-time estimator is far superior to the traditional discrete-time approach to TE estimation on spike trains. For a start, unlike the discrete-time estimator, it is consistent. That is, in the limit of infinite data, it will converge to the true value of the TE. It was also shown to have much preferable bias and convergence properties. Most significantly, perhaps, this new estimator utilises the interspike intervals to efficiently represent the history embeddings $\mathbf{x}_{<x_i}$ and $\mathbf{y}_{<x_i}$ in estimating the relevant conditional spike rates in (*Lizier, 2013*). This then allows for the application of the highly effective nearest-neighbour family of information-theoretic estimators (*Kozachenko and Leonenko, 1987*; *Kraskov et al., 2004*), which bring estimation efficiency, bias correction, and together with their application to interspike intervals enable capture of long timescale dependencies.

This is in contrast to the traditional discrete-time estimator which uses the presence or absence of spikes in time bins as its history embeddings (it sometimes also uses the number of spikes occurring in a bin). In order to avoid the dimensionality of the estimation problem becoming sufficiently large so as to render estimation infeasible, only a small number of bins can be used in these embeddings. Indeed, to the best of the authors' knowledge, all previous applications of the discrete-time TE estimator to spiking data from cell cultures used only a single bin in their history embeddings. The bin widths used in those studies were 40 s (*Nigam et al., 2016*), 0.3 ms (*Garofalo et al., 2009*), and 1 ms (*Shimono and Beggs, 2015*; *Kajiwara et al., 2020*). Some studies chose to examine the TE values produced by multiple different bin widths, specifically, 0.6 ms and 100 ms (*Matsuda et al., 2013*), 1.6 ms and 3.5 ms (*Timme et al., 2016*), and 10 different widths ranging from 1 ms to 750 ms (*Timme et al., 2014*). Specifically those studies demonstrated the unfortunate high sensitivity of the discrete-time TE estimator to the bin width parameter. In the instances where narrow (<5 ms) bins were used, only a very narrow slice of history is being considered in the estimation of the history-conditional spike rate. This is problematic as it is known that correlations in spike trains exist over distances of (at least) hundreds of milliseconds (*Aldridge and Gilman, 1991*; *Rudelt et al., 2021*). Conversely, in the instances where broad (>5 ms) bins were used, relationships occurring on fine timescales will be completely missed. This is significant given that it is established that correlations at the millisecond and sub-millisecond scale play a role in neural function (*Nemenman et al., 2008*; *Kayser et al., 2010*; *Sober et al., 2018*; *Garcia-Lazaro et al., 2013*). In other words, previous applications of TE to electrophysiological data from cell cultures either captured some correlations occurring with fine temporal

**Table 6.** The parameter values used in the continuous-time transfer entropy (TE) estimator. A complete description of these parameters, along with analysis and discussion of their effects, can be found in *Shorten et al., 2021*.

| Parameter | Description | Value |
|---|---|---|
| $N_X$ | Number of spikes in the target spike train | Varied (see text) |
| $l_X$ | Number of interspike intervals in target history embeddings | 4 |
| $l_Y$ | Number of interspike intervals in source history embeddings | 2 |
| $k_{\text{global}}$ | Number of nearest neighbours to find in the initial search | 10 |
| $k_{\text{perm}}$ | Number of nearest neighbours to consider during surrogate generation | 10 |
| $N_U$ | Number of random samples of histories at non-spiking points in time | $50N_X$ |
| $N_{U,\text{surrogates}}$ | Number of random samples of histories at non-spiking points in time used for surrogate generation | $10N_X$ |
| $N_{\text{surrogates}}$ | Number of surrogates to generate for each node pair | 100 |

precision or they captured relationships occurring over larger intervals, but never both simultaneously. This can be contrasted with the interspike interval history representation used in this study. To take a concrete example, for the recording on day 24 of culture 1-3, the average interspike interval was 0.71 s. This implies that the target history embeddings (composed of four interspike intervals) on average extended over a period of 2.84 s and the source history embeddings (composed of two interspike intervals) on average extended over a period of 1.42 s. This is despite the fact that our history representations retain the precision of the raw data (40 s) and the ability to measure relationships on this scale where they are relevant (via the underlying nearest-neighbour estimators).

The parameters used with this estimator are shown in *Table 6*. The values of $k_{\text{global}}$ and $k_{\text{perm}}$ were chosen because, in previous work (*Shorten et al., 2021*), similar values were found to facilitate stable performance of the estimator. The high values of $N_U$ and $N_{U,\text{surrogates}}$ were chosen so that histories during bursting periods could be adequately sampled. These two parameters refer to sample points placed randomly in the spike train, at which history embeddings are sampled. As the periods of bursting comprise a relatively small fraction of the total recording time, many samples need to be placed in order to achieve a good sample of histories potentially observed during these periods. The choice of embedding lengths is discussed in the section 'Selection of embedding lengths', and the choice of $N_{\text{surrogates}}$ is discussed in the section 'Significance testing of TE values'.

Instead of selecting a single number of target spikes $N_X$ to include in the analysis, we chose to include all the spikes that occurred within the first hour of recording time. The reason for doing this was that the spike rates varied by orders of magnitude between the electrodes. This meant that fixing the number of target spikes would result in the source spikes being severely undersampled in cases where the target spike rate was much higher than the source spike rate. When using 1 hr of recording time, among the main cultures the smallest number of spikes per electrode was 481, the maximum was 69,627, and the median was 17,399.

## Selection of embedding lengths

The target embedding lengths were determined by adapting the technique (*Erten et al., 2017*; *Novelli et al., 2019*) extending (*Garland et al., 2016*) of maximising the bias-corrected active information storage (AIS) (*Lizier et al., 2012*) over different target embedding lengths for a given target. Our adaptations sought to select a consensus embedding parameter for all targets on all trials to avoid different bias properties due to different parameters across targets and trials, in a similar fashion to *Hansen et al., 2021*. As such, our approach determines a target embedding length $l_X$ which maximises the *average* bias-corrected AIS across all electrodes using one representative recording

**Table 7.** Summary statistics for the active information storage (AIS) values estimated at different target embedding lengths $l_X$.

These were estimated across all electrodes of a representative recording (day 23 of culture 1-3). The p-values shown in the fourth column are associated with the null hypothesis that the mean AIS at the given $l_X$ is equal to the mean AIS at $l_X - 1$.

| $l_X$ | Mean AIS | SD | p-Value |
|---|---|---|---|
| 1 | 7.73 | 4.71 | – |
| 2 | 8.27 | 4.97 | $3.0 \times 10^{-19}$ |
| 3 | 8.41 | 5.08 | $5.8 \times 10^{-8}$ |
| 4 | 8.44 | 5.11 | $2.7 \times 10^{-4}$ |
| 5 | 8.43 | 5.12 | 0.85 |

(selected as day 23 of culture 1-3). To estimate AIS within the continuous-time framework (*Spinney and Lizier, 2018*) for this purpose, we estimated the difference between the second KL divergence of Equation 10 of *Shorten et al., 2021* and the mean firing rate of the target. These estimates contain inherent bias correction as per the TE estimator itself. Moreover, the mean of surrogate values was subtracted to further reduce the bias. The embedding length $l_X$ was continuously increased so long as each subsequent embedding produced a statistically significant (at the p<0.05 level) increase in the average AIS across the electrodes. The resulting mean AIS values (along with standard deviations) and p-values are shown in *Table 7*. We found that every increase in $l_X$ up to 4 produced a statistically significant increase in the mean AIS. The increase from 4 to 5 produced a non-significant decrease in the mean AIS and so $l_X$ was set to 4.

With the target embedding length determined, we set about similarly determining a consensus source embedding length $l_Y$ by estimating the TE between all directed electrode pairs on the same representative recording for different values of $l_Y$. Each estimate also had the mean of the surrogate population subtracted to reduce its bias (see section 'Significance testing of TE values').

The embedding length was continuously increased so long as each subsequent embedding produced a statistically significant (at the p<0.05 level) increase in the average TE across all electrode pairs. The resulting mean TE values (along with standard deviations) and p-values are shown in *Table 8*. We found that increasing $l_Y$ from 1 to 2 produced a statistically significant increase in the mean TE. However, increasing $l_Y$ from 2 to 3 produced a non-significant decrease in the mean TE. As such, we set $l_Y$ to 2.

## Significance testing of TE values

In constructing the directed functional networks displayed in *Figure 2a*, we tested whether the estimated TE between each source-target pair was statistically different from the distribution of TEs under the null hypothesis of conditional independence of the target from the source (i.e. TE consistent with zero). Significance testing for TE in this way is performed by constructing a population of surrogate time-series or history embeddings that conform to the null hypothesis of zero TE (*Novelli et al., 2019*; *Wollstadt et al., 2019*; *Novelli and Lizier, 2021*). We then estimate the TE on each of

**Table 8.** Summary statistics for the transfer entropy (TE) values estimated at different source embedding lengths $l_Y$.

These were estimated between all electrodes of a representative recording (day 23 of culture 1-–3). The p-values shown in the fourth column are associated with the null hypothesis that the mean TE at the given $l_Y$ is equal to the mean TE at $l_Y - 1$.

| $l_Y$ | Mean TE | SD | p-Value |
|---|---|---|---|
| 1 | 0.031 | 0.043 | – |
| 2 | 0.058 | 0.056 | 0.0 |
| 3 | 0.057 | 0.069 | 0.84 |

these surrogates to generate a null distribution of TE. Specifically, we generate the surrogates and compute their TEs according to the method associated with the continuous-time spiking TE estimator (*Shorten et al., 2021*) and using the parameters shown in *Table 6*. One small change was made to that surrogate generation method: instead of laying out the $N_{U,\text{surrogates}}$ sample points randomly uniformly, we placed each one at an existing target spike, with the addition of uniform noise on the interval $[-240\,\text{ms}, 240\,\text{ms}]$. This was to ensure that these points adequately sampled the incredibly dense burst regions.

With the surrogate TE distribution constructed, the resulting p-value for our TE estimate can be computed by counting the proportion of these surrogate TEs that are greater than or equal to the original estimate. Here, we seek to compare significance against a threshold of $\alpha < 0.01$. We chose this lower threshold as false positives are generally considered more damaging than false negatives when applying network inference to neuroscientific data (*Zalesky et al., 2016*). We also applied a Bonferroni correction (*Miller, 2012*) to all the significance tests done on a given recording. Given that there are 59 electrodes in the recordings, 3422 tests were performed in each recording. This meant that, once the Bonferroni correction was included, the significance threshold dropped to $\text{p}<2.9 \times 10^{-6}$. Comparing against such a low significance threshold would require an infeasible number of surrogates for the many pairs within each recording, if computing the p-value by counting as above. Instead, we assume that the null TE distribution is Gaussian and compute the p-value for our TE estimate using the CDF of the Gaussian distribution fitted from 100 surrogates (e.g. as per *Lizier et al., 2011*). Specifically, the p-value reports the probability that a TE estimate on history embeddings conforming to the null hypothesis of zero TE being greater than or equal to our original estimated TE value. If this p-value is below the threshold, then the null hypothesis is rejected and we conclude that there is a statistically significant information flow between the electrodes.

## Analysis of population bursts

A common family of methods for extracting periods of bursting activity from spike-train recordings examines the length of adjacent interspike intervals. The period spanned by these intervals is designated a burst if some summary statistic of the intervals (e.g. their sum or maximum) is below a certain threshold (*Kaneoke and Vitek, 1996*; *Wagenaar et al., 2005*; *Wagenaar et al., 2006b*; *Selinger et al., 2007*; *Bakkum et al., 2013*). In order to detect single-electrode as well as population-wide bursts, we implement such an approach here.

We first determine the start and end points of the bursts of each individual electrode. The locations of the population bursts were subsequently determined using the results of this per-electrode analysis.

The method for determining the times during which an individual electrode was bursting proceeded as follows: the spikes were moved through sequentially. If the interval between a given spike and the second most recent historic spike for that electrode was less than $\alpha$, then, if the electrode was not already in a burst, it was deemed to have a burst starting at the $k$th most recent historic spike. A burst was taken to continue until an interspike interval greater than $a * \alpha$ was encountered. If such an interval was encountered, then the end of the burst was designated as the timestamp of the earlier of the two spikes forming the interval.

The starts and ends of population bursts were similarly determined by moving through the time series in a sequential fashion. If the population was not already designated to be in a burst, but the number of electrodes currently bursting was greater than the threshold $\beta$, then a burst start position was set at the point this threshold was crossed. Conversely, if the electrode was already designated to be in a burst and the number of individual electrodes currently bursting dropped below the threshold $\gamma$ ($\gamma < \beta$), then a burst stop position was set at the point this threshold was crossed.

In this article, we always made use of the parameters $k = 2, \alpha = \frac{1}{2\bar{\lambda}}, a = 3, \beta = 15$ and $\gamma = 10$, where $\bar{\lambda}$ is the average spike rate. These parameters were chosen by trial-and-error combined with visual inspection of the resulting inferred burst positions. The results of this scheme showed low sensitivity to the choice of these parameters.

For the simulated network dynamics, we used the parameters $k = 1, \alpha = \frac{1}{2\bar{\lambda}}, a = 1.5, \beta = 2$ and $\gamma = 1$. These parameters were found to better suit the stereotyped dynamics of the simulated networks.

## Estimation of burst-local TE

The information dynamics framework provides us with the unique ability to analyse information processing locally in time (*Lizier et al., 2008*; *Lizier, 2013*; *Lizier, 2014*). We make use of that ability here to allow us to specifically examine the information flows during the important period of population bursts. The TE estimator which we are employing here (*Shorten et al., 2021*) sums contributions from each spike in the target spike train. It then divides this total by the time length of the target spike train that is being examined. In order to estimate the burst-local TE, we simply sum the contributions from the target spikes where those spikes occurred during a population burst. We then normalise by the number of such spikes, providing us with a burst-local TE estimate in units of nats per spike, instead of nats per second. Note that the burst-local TE is different to the approach of *Stetter et al., 2012*, who extracted the bursting activity prior to any analysis, rendering a TE conditioned on bursting occurring. Specifically, in contrast to the burst-local TE, in their work the non-spiking activity is ignored for the purposes of estimating the log densities.

## Code availability

Scripts for performing the analysis in this article can be found at bitbucket.org/dpshorten/cell_cultures (*Shorten, 2022*; copy archived at swh:1:rev:8ee5e519da5cb90590865e9a692b96ad7e68a69e).

## Acknowledgements

JL was supported through the Australian Research Council DECRA grant DE160100630 (https://www.arc.gov.au/grants/discovery-program/discovery-early-career-researcher-award-decra) and The University of Sydney Research Accelerator (SOAR) prize program (https://www.sydney.edu.au/research/our-researchers/sydney-research-accelerator-prizes.html). The authors acknowledge the technical assistance provided by the Sydney Informatics Hub, a Core Research Facility of the University of Sydney. In particular, the analysis presented in this work made use of the Artemis HPC cluster.

## Additional information

### Funding

| Funder | Grant reference number | Author |
|---|---|---|
| Australian Research Council | DE160100630 | Joseph T Lizier |
| University of Sydney | SOAR Fellowship | Joseph T Lizier |
| Deutsche Forschungsgemeinschaft | SFB 1528 | Viola Priesemann |

The funders had no role in study design, data collection and interpretation, or the decision to submit the work for publication.

### Author contributions

David P Shorten, Conceptualization, Investigation, Methodology, Software, Visualization, Writing - original draft, Writing - review and editing; Viola Priesemann, Michael Wibral, Conceptualization, Writing - review and editing; Joseph T Lizier, Conceptualization, Funding acquisition, Methodology, Project administration, Resources, Supervision, Writing - review and editing

### Author ORCIDs

David P Shorten http://orcid.org/0000-0003-2412-4705
Viola Priesemann http://orcid.org/0000-0001-8905-5873
Michael Wibral http://orcid.org/0000-0001-8010-5862
Joseph T Lizier http://orcid.org/0000-0002-9910-8972

### Decision letter and Author response

Decision letter https://doi.org/10.7554/eLife.74651.sa1
Author response https://doi.org/10.7554/eLife.74651.sa2

## Additional files

### Supplementary files
• Transparent reporting form

### Data availability
This work made use of a publicly available dataset which can be found at: http://neurodatasharing.bme.gatech.edu/development-data/html/index.html. Analysis scripts are available at: https://bitbucket.org/dpshorten/cell_cultures (copy archived at swh:1:rev:8ee5e519da5cb90590865e9a692b96ad7e68a69e).

The following previously published dataset was used:

| Author(s) | Year | Dataset title | Dataset URL | Database and Identifier |
|---|---|---|---|---|
| Pine J, Potter S | 2006 | Network activity of developing cortical cultures in vitro | http://neurodatasharing.bme.gatech.edu/development-data/html/index.html | neurodatasharing, development-data |

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

## Appendix 1

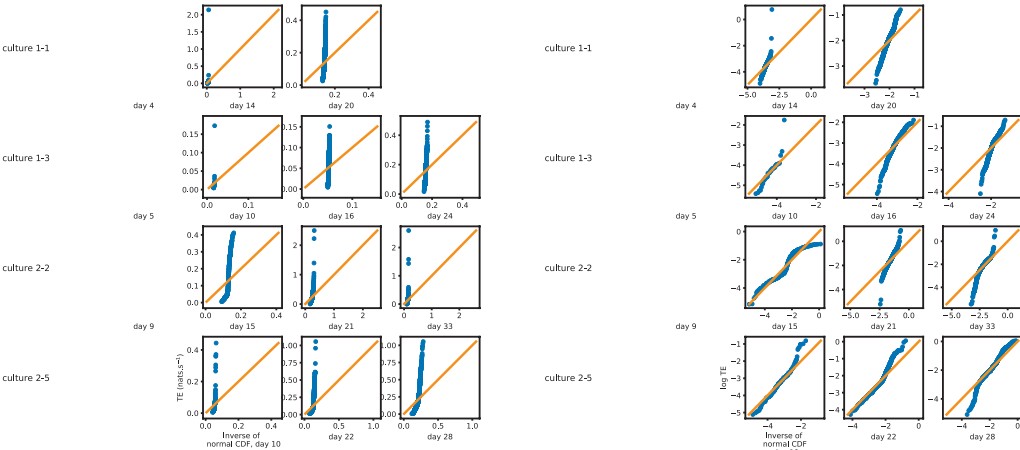

(a)   QQ plots of TE values against the normal distribution.

(b)   QQ plots of log TE values against the normal distribution.

**Appendix 1—figure 1.** Quantile-quantile (QQ) plots (*Gibbons and Chakraborti, 2020*) of the nonzero estimated transfer entropy (TE) values against normal (**a**) and log-normal (**b**) distributions, respectively. The $y$ axis shows estimated TE values (or their logarithm), whereas the $x$ axis shows the value of the normal distribution at the same quantile. The solid orange line shows the line $y = x$. If the data is drawn from the distribution against which it is being plotted, then the blue marks will sit along this line. We observe that the distributions of TE values deviate substantially from both normal and log-normal distributions in all recordings analysed.

**Appendix 1—table 1.** p-Values for the Shapiro–Wilk test (*Shapiro and Wilk, 1965*) of normality for the distribution of transfer entropy (TE) values estimated in each recording.
Only the statistically significant TE values are included in these tests. Recordings for which there were no statistically significant values estimated are left blank. These p-values represent the probability that the associated test statistic is more extreme than that calculated on the estimated TE values, under the null hypothesis that these values are normally distributed. For any reasonable choice of p cutoff value, the null hypothesis is rejected in all recordings.

| Culture 1-1 | Day 4 | Day 14 | Day 20 | |
|---|---|---|---|---|
| | – | $9.8 \times 10^{-45}$ | $4.2 \times 10^{-35}$ | |
| Culture 1-3 | Day 5 | Day 10 | Day 16 | Day 24 |
| | $4.4 \times 10^{-13}$ | $4.7 \times 10^{-24}$ | $2.7 \times 10^{-36}$ | $1.0 \times 10^{-36}$ |
| Culture 2-2 | Day 9 | Day 15 | Day 21 | Day 33 |
| | $7.9 \times 10^{-6}$ | $1.6 \times 10^{-28}$ | $2.6 \times 10^{-35}$ | $9.5 \times 10^{-38}$ |
| Culture 2-5 | Day 4 | Day 10 | Day 22 | Day 28 |
| | – | $7.5 \times 10^{-10}$ | $2.4 \times 10^{-28}$ | $3.7 \times 10^{-29}$ |

**Appendix 1—table 2.** p-Values for the Shapiro–Wilk test (*Shapiro and Wilk, 1965*) of normality for the distribution of transfer entropy (TE) values estimated in each recording.
Only the statistically significant TE values are included in these tests. Recordings for which there were no statistically significant values estimated are left blank. These p-values represent the probability that the associated test statistic is more extreme than that calculated on the estimated TE values, under the null hypothesis that these values are normally distributed. For any reasonable choice of p cutoff value, the null hypothesis is rejected in all recordings.

| Culture 1-1 | Day 4 | Day 14 | Day 20 |
|---|---|---|---|

*Appendix 1—table 2 Continued on next page*

*Appendix 1—table 2 Continued*

| | | | |
|---|---|---|---|
| – | 0 | $3.3\times10^{-33}$ | |
| **Culture 1-2** | Day 6 | Day 11 | Day 17 |
| – | | $2.3\times10^{-21}$ | $3.8\times10^{-43}$ |
| **Culture 1-3** | Day 5 | Day 10 | Day 16 | Day 24 |
| – | | $6.4\times10^{-13}$ | $3.3\times10^{-14}$ | $5.3\times10^{-14}$ |
| **Culture 1-4** | Day 8 | Day 13 | Day 19 |
| – | | $2.5\times10^{-28}$ | $2.1\times10^{-31}$ |
| **Culture 1-5** | Day 7 | Day 12 | Day 18 |
| $8.4\times10^{-2}$ | $1\times10^{-35}$ | $1.5\times10^{-12}$ | |
| **Culture 2-1** | Day 14 | Day 32 |
| $2.8\times10^{-1}$ | 0 | | |
| **Culture 2-2** | Day 9 | Day 15 | Day 21 | Day 33 |
| – | | $1.4\times10^{-19}$ | $7.1\times10^{-44}$ | 0 |
| **Culture 2-3** | Day 6 | Day 12 | Day 24 |
| – | | $2.6\times10^{-1}$ | $1.4\times10^{-45}$ |
| **Culture 2-4** | Day 3 | Day 5 | Day 11 |
| $3.5\times10^{-12}$ | – | $1.0\times10^{-2}$ | |
| **Culture 2-5** | Day 4 | Day 10 | Day 22 | Day 28 |
| – | | $1.2\times10^{-21}$ | $3.5\times10^{-40}$ | $1.2\times10^{-37}$ |
| **Culture 2-6** | Day 7 | Day 13 | Day 31 |
| $1.0\times10^{-1}$ | – | $1.9\times10^{-19}$ | – |

**Appendix 1—table 3.** p-Values for the Shapiro–Wilk test (**Shapiro and Wilk, 1965**) of log-normality for the distribution of transfer entropy (TE) values estimated in each recording.

Only the statistically significant TE values are included in these tests. Recordings for which there were no statistically significant values estimated are left blank. These p-values represent the probability that the associated test statistic is more extreme than that calculated on the logarithms of the estimated TE values, under the null hypothesis that these values are normally distributed. For any reasonable choice of p cutoff value, the null hypothesis is rejected in all recordings (apart from those with very few significant TE values). It is interesting to note that the p-values are often smaller on later days, despite the Q-Q plots in *Appendix 1—figure 1*, suggesting the distribution is closer to log-normal. This is probably due to there being many more statistically significant TE values on these later days (see *Table 2*).

| | | | | |
|---|---|---|---|---|
| **Culture 1-1** | Day 4 | Day 14 | Day 20 | |
| – | | $3.1\times10^{-21}$ | $2.9\times10^{-10}$ | |
| **Culture 1-2** | Day 6 | Day 11 | Day 17 | |
| – | | $3.2\times10^{-14}$ | $1.6\times10^{-22}$ | |
| **Culture 1-3** | Day 5 | Day 10 | Day 16 | Day 24 |
| – | | $1.3\times10^{-4}$ | $3.0\times10^{-13}$ | $1.3\times10^{-22}$ |
| **Culture 1-4** | Day 8 | Day 13 | Day 19 | |
| – | | $3.0\times10^{-15}$ | $2.4\times10^{-7}$ | |
| **Culture 1-5** | Day 7 | Day 12 | Day 18 | |
| $3.3\times10^{-2}$ | $7.8\times10^{-24}$ | $2.0\times10^{-4}$ | | |

*Appendix 1—table 3 Continued on next page*

Appendix 1—table 3 Continued

| Culture 2-1 | Day 14 | Day 32 | | |
|---|---|---|---|---|
| | $9.7 \times 10^{-1}$ | $3.6 \times 10^{-22}$ | | |
| Culture 2-2 | Day 9 | Day 15 | Day 21 | Day 33 |
| | – | $1.8 \times 10^{-12}$ | $3.6 \times 10^{-14}$ | $5.8 \times 10^{-29}$ |
| Culture 2-3 | Day 6 | Day 12 | Day 24 | |
| | – | $6.1 \times 10^{-2}$ | $1.78 \times 10^{-7}$ | |
| Culture 2-4 | Day 3 | Day 5 | Day 11 | |
| | $9.8 \times 10^{-13}$ | – | $5.1 \times 10^{-2}$ | |
| Culture 2-5 | day 4 | day 10 | Day 22 | Day 28 |
| | - | $1.2 \times 10^{-3}$ | $1.1 \times 10^{-16}$ | $2.4 \times 10^{-14}$ |
| Culture 2-6 | day 7 | day 13 | Day 31 | |
| | $7.4 \times 10^{-1}$ | - | $1.9 \times 10^{-14}$ | – |

Previous studies have placed an emphasis on the observation of log-normal distributions of TE values in in vitro cultures of neurons (*Shimono and Beggs, 2015*; *Nigam et al., 2016*). As such, we analysed the distribution of the nonzero (statistically significant) estimated TE values in each individual recording.

*Figure 1* shows histograms as well as probability density functions estimated by a kernel density estimator (KDE) of the nonzero TE values for each recording. From these plots, we can see that the distributions of TE values exhibit a clear right (positive) skew. In order to ascertain how well the estimated TE values were described by a log-normal distribution, we constructed quantile-quantile (QQ) plots (*Gibbons and Chakraborti, 2020*) for the TE values against the log-normal distribution in *Appendix 1—figure 1*. In all recordings, the plotted points deviate from the line $y = x$, indicating that the data is not well described by a log-normal distribution. However, this deviation appears only slight for some recordings, most notably days 22 and 28 of culture 2-5. We also perform Shapiro–Wilk tests (*Shapiro and Wilk, 1965*) for log-normality, the resulting p-values are displayed in *Appendix 1—table 3*. The p-values for every recording are incredibly low, meaning that we reject the null hypothesis of a log-normal distribution in every case.

Given that the distributions of the TE values were not well described by a log-normal distribution, we investigated the alternative that they could be described by a normal distribution. *Appendix 1—figure 1* displays QQ plots (*Gibbons and Chakraborti, 2020*) for the TE values against the normal distribution. In all recordings, the plotted points deviate substantially from the line $y = x$, indicating that the data is poorly described by a normal distribution. We also perform Shapiro–Wilk tests (*Shapiro and Wilk, 1965*) for normality, the resulting p-values are displayed in *Appendix 1—table 1*. The p-values for every recording are incredibly low, meaning that we reject the null hypothesis of a normal distribution in every case.

These results contrast with observation of log-normal distributions of TE values in in vitro cultures of neurons (*Shimono and Beggs, 2015*; *Nigam et al., 2016*). The difference may be due to the use of continuous-time estimator here in contrast to the discrete-time estimator used in previous studies. This estimator is more faithful to capturing the true underlying TE for spike trains (as per *Shorten et al., 2021*); however, it may be that the combination of the discrete-time estimator and use of only a single previous time-bin – in specifically *not* representing history dependence well – align more strongly with the component of the statistical relationship that follows a log-normal distribution. It is also possible that log-normal distributions of TE emerge later in development and are simply not yet present in the early developmental stages observed here (noting that the fit to a log-normal distribution seems to improve for later DIV in *Appendix 1—figure 1*).

# Appendix 2

## Plots for early lock-in of incoming TE

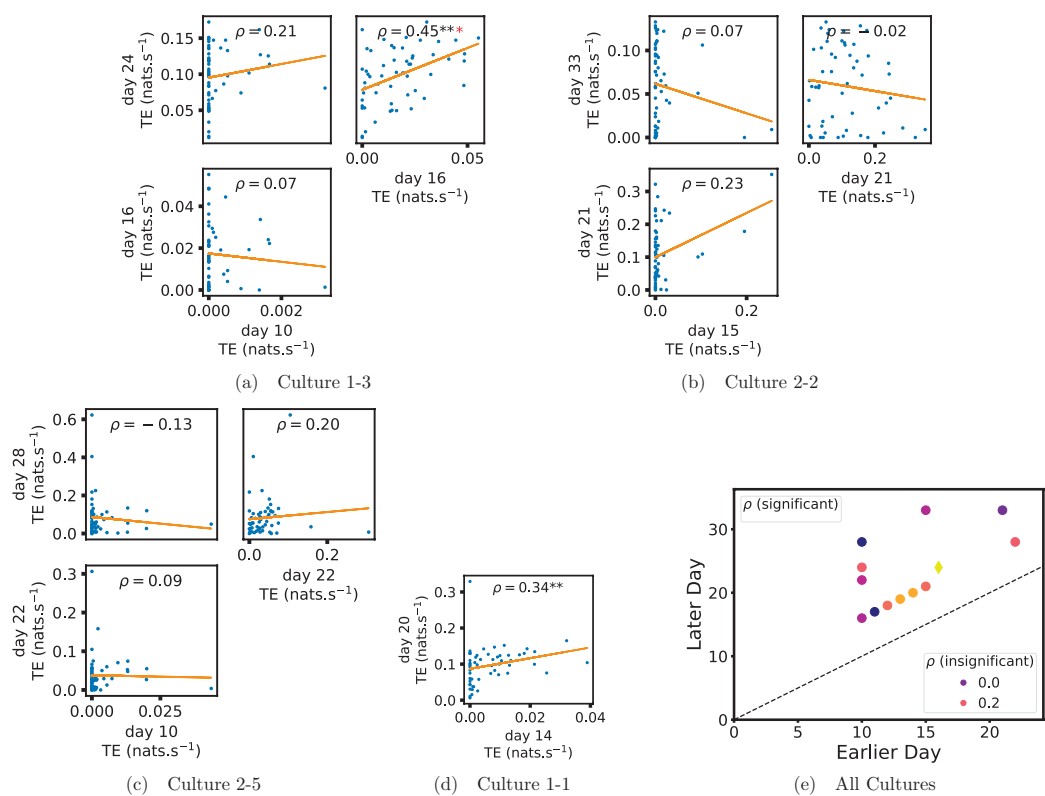

**Appendix 2—figure 1.** Plots investigating the relationship between the inward information flow from a given node over different days of development. (**a**–**d**) show scatter plots between all pairs of days for each culture (excluding days with zero significant transfer entropy [TE] values). Specifically, in each scatter plot, the $x$ value of a given point is the average inward TE from the associated node on an earlier day and the $y$ value of that same point is the total outgoing TE from the same node but on a later day. The days in question are shown on the bottom and sides of the grids of scatter plots. The orange line shows the ordinary least squares regression. The Spearman correlation ($\rho$) between the outgoing TE values on the two days is displayed in each plot. Values of $\rho$ significant at the 0.05 level are designated with an asterisk and those significant at the 0.01 level are designated with a double asterisk. A Bonferroni correction for multiple comparisons was used. (**e**) shows all recording day pairs for all cultures (where the pairs are always from the same culture) and the associated Spearman correlation between the outward TEs of nodes across this pair of recording days. Diamonds indicate significance at $p<0.05$, with Bonferroni correction.

# Appendix 3

## Extra cultures

**Appendix 3—table 1.** Mean transfer entropy (TE) in nats per second between every source-target pair for the additional cultures.

| Culture 1-2 | Day 6 | Day 11 | Day 17 |
|---|---|---|---|
| | 0 | $6.6\times10^{-4}$ | 0.023 |
| Culture 1-4 | Day 8 | Day 13 | Day 19 |
| | 0 | 0.017 | 0.040 |
| Culture 1-5 | Day 7 | Day 12 | Day 18 |
| | $5.6\times10^{-5}$ | $6.6\times10^{-3}$ | 0.016 |
| Culture 2-1 | Day 14 | Day 32 | |
| | $2.9\times10^{-6}$ | 0.028 | |
| Culture 2-3 | Day 6 | Day 12 | Day 24 |
| | 0 | $2.0\times10^{-5}$ | 0.075 |
| Culture 2-4 | Day 3 | Day 5 | Day 11 |
| | $6.4\times10^{-3}$ | 0 | $5.3\times10^{-5}$ |
| Culture 2-6 | Day 7 | Day 13 | Day 31 |
| | 0 | 0 | 0.061 |

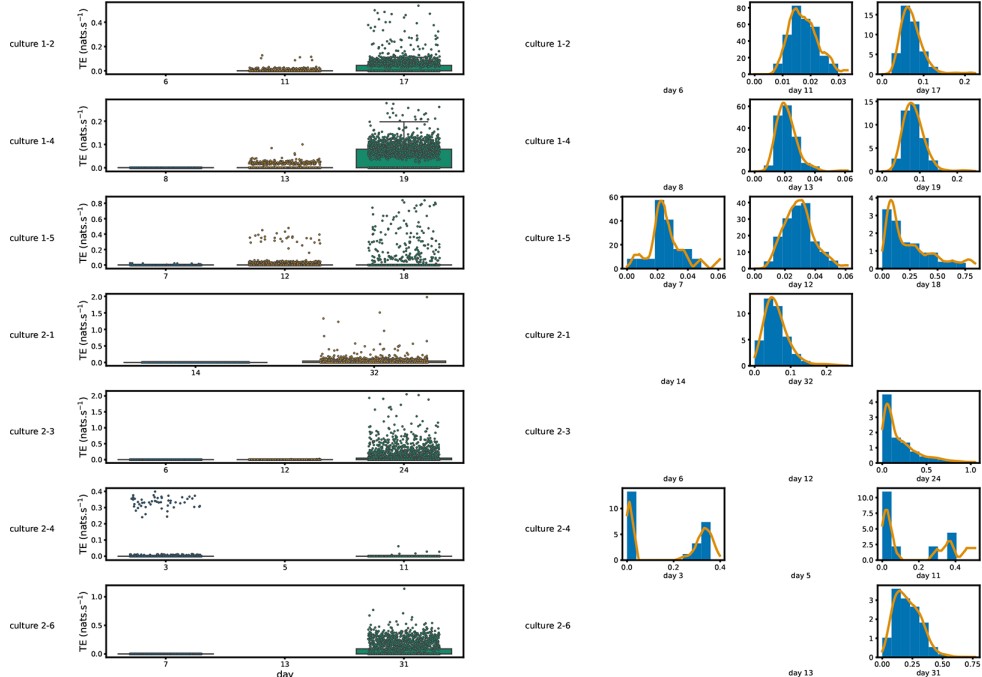

(a)  Scatters and boxplots of TE values.      (b)  Histograms and kernel density estimates of TE values.

**Appendix 3—figure 1.** Identical plots to those shown in *Figure 1*, but showing the cultures left out of that plot for space considerations. (**a**) Scatters of the TE values are overlaid on box plots. The box plots show the quartiles and the median (values greater than 10 standard deviationSDs from the mean have been removed from both the box and scatter plots as outliers). (**b**) Density estimates of the nonzero (statistically significant) TE distribution on top of a histogram. The densities are estimated using a Gaussian kernel. The histogram bin width and kernel

histogram are both 10% of the data range. Recordings with fewer than 10 statistically significant TE values are excluded.

**Appendix 3—table 2.** Displays the same information as *Figure 2*, but for the additional cultures.

| Culture 1-2 | Day 6 | Day 11 | Day 17 |
| --- | --- | --- | --- |
| | 0 | 105 | 860 |
| Culture 1-4 | Day 8 | Day 13 | Day 19 |
| | 1 | 214 | 1457 |
| Culture 1-5 | Day 7 | Day 12 | Day 18 |
| | 21 | 375 | 195 |
| Culture 2-1 | Day 14 | Day 32 | |
| | 5 | 1165 | |
| Culture 2-3 | Day 6 | Day 12 | Day 24 |
| | 2 | 9 | 1000 |
| Culture 2-4 | Day 3 | Day 5 | Day 11 |
| | 97 | 0 | 11 |
| Culture 2-6 | Day 7 | Day 13 | Day 31 |
| | 9 | 0 | 873 |

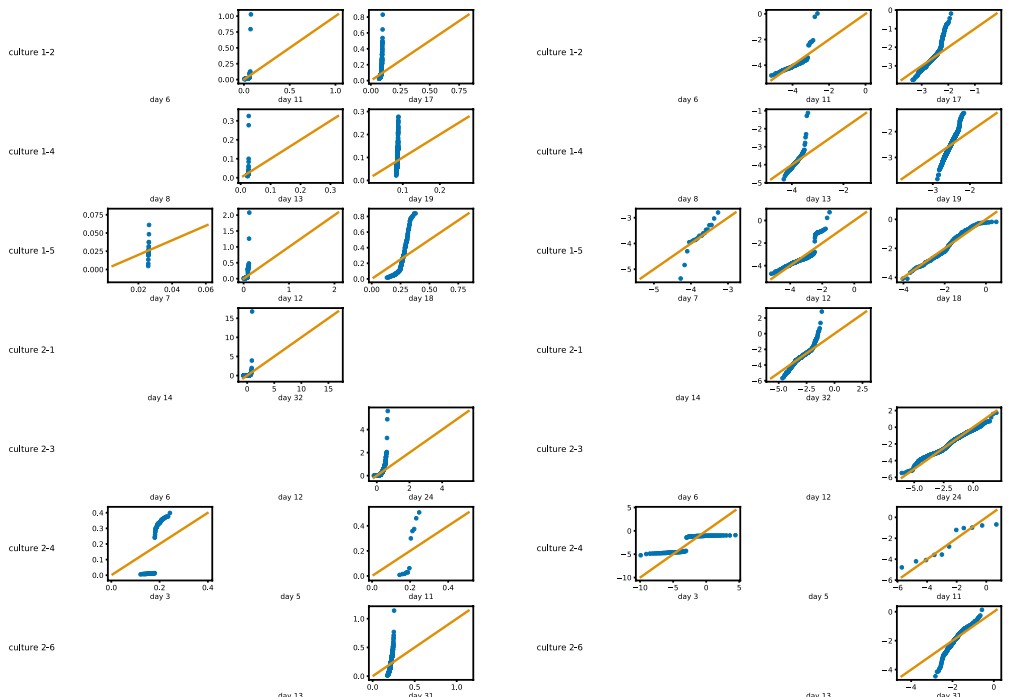

(a) QQ plots of TE values against the normal distribution.     (b) QQ plots of log TE values against the normal distribution.

**Appendix 3—figure 2.** Identical plots to those shown in *Appendix 2—figure 1*, but for the additional cultures. The $y$ axis shows estimated TE values (or their logarithm), whereas the axis shows the value of the normal distribution at the same quantile. The solid orange line shows the line $y = x$. If the data is drawn from the *Appendix 3—figure 2 continued on next page*

*Appendix 3—figure 2 continued*
distribution against which it is being plotted, then the blue marks will sit along this line. We observe that the distributions of TE values deviate substantially from both normal and log-normal distributions in all recordings analysed.

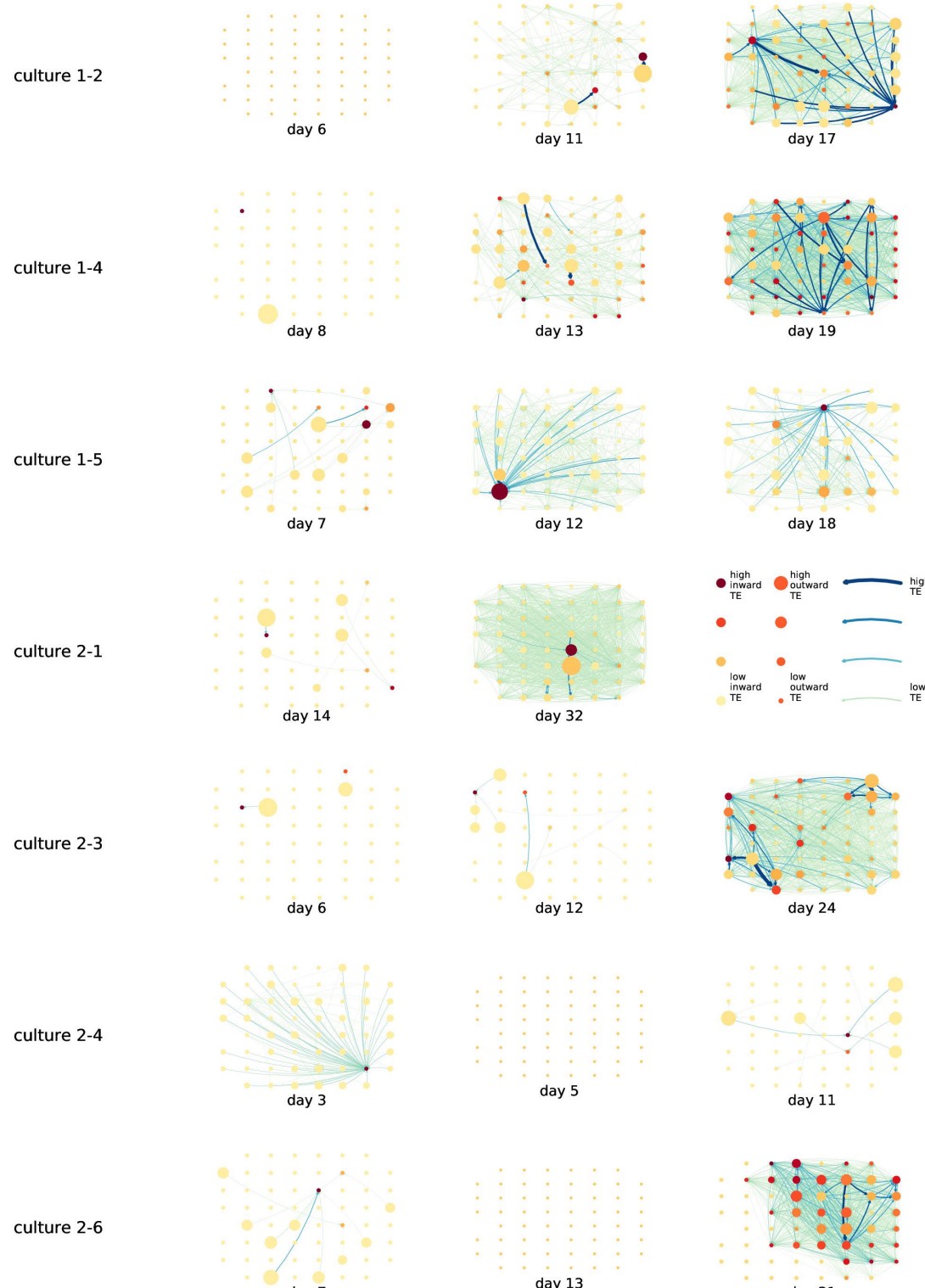

**Appendix 3—figure 3.** Identical plots to those in *Figure 2*, but for the additional cultures.

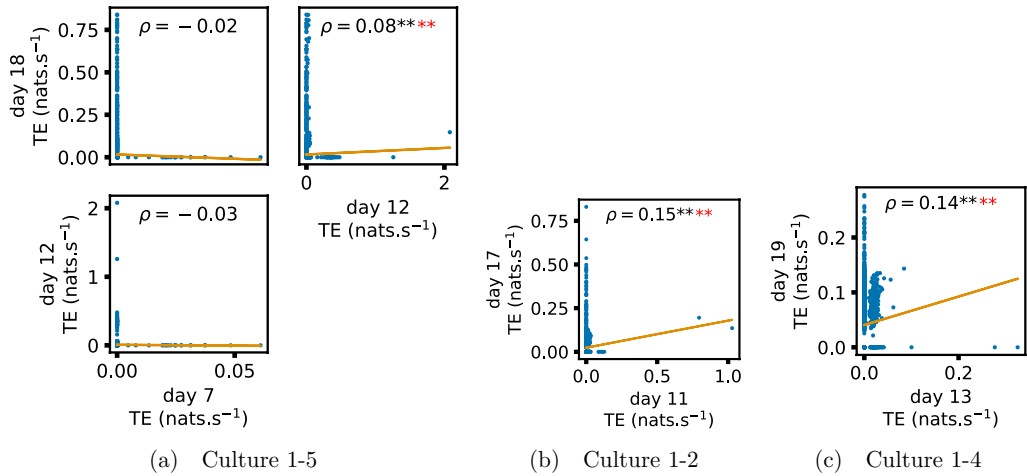

(a) Culture 1-5  (b) Culture 1-2  (c) Culture 1-4

**Appendix 3—figure 4.** Identical plots to those in *Figure 3*, but for the additional cultures. (a) Contains plots for culture 1-5, (b) contains plots for culture 1-2 and (c) contains plots for culture 1-4.

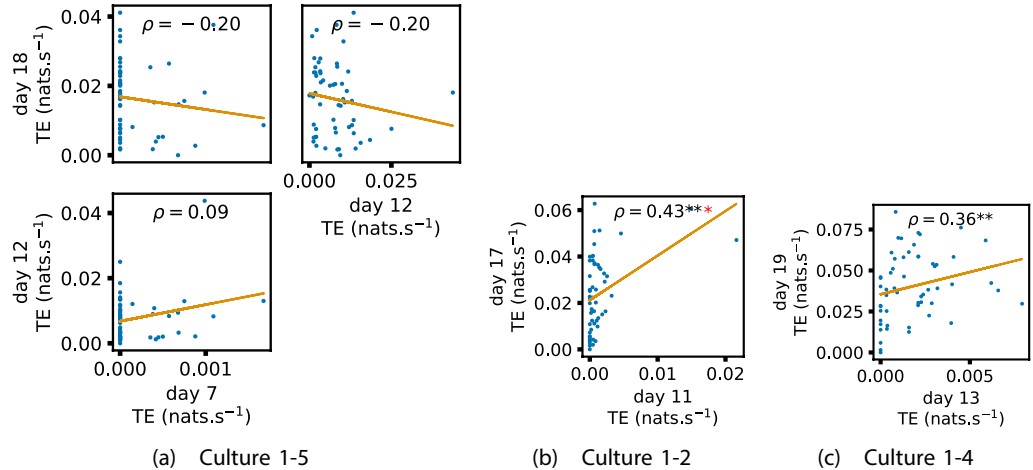

(a) Culture 1-5  (b) Culture 1-2  (c) Culture 1-4

**Appendix 3—figure 5.** Identical plots to those in *Figure 4*, but for the additional cultures. (**a**) Contains plots for culture 1-5, (**b**) contains plots for culture 1-2 and (**c**) contains plots for culture 1-4.

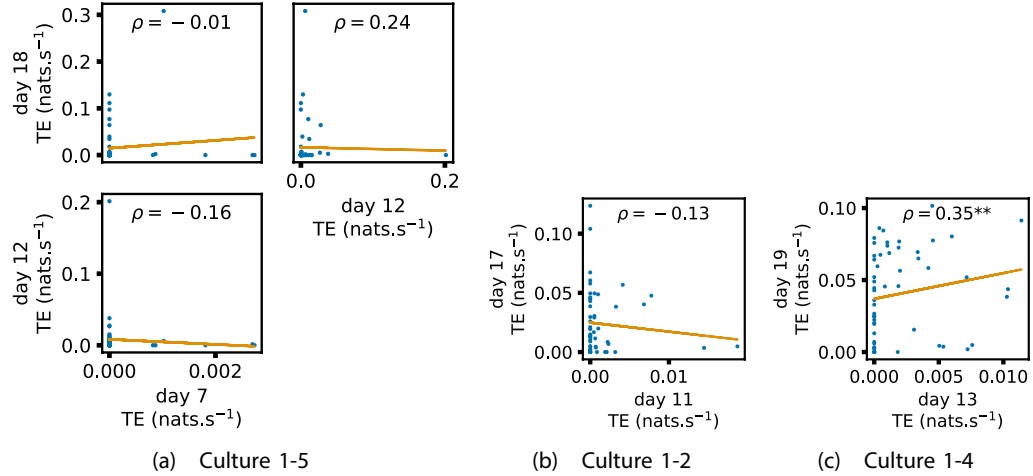

**Appendix 3—figure 6.** Identical plots to those in *Appendix 2—figure 1*, but for the additional cultures. (**a**) Contains plots for culture 1-5. (**b**) Contains plots for culture 1-2. (**c**) Contains plots for culture 1-4.

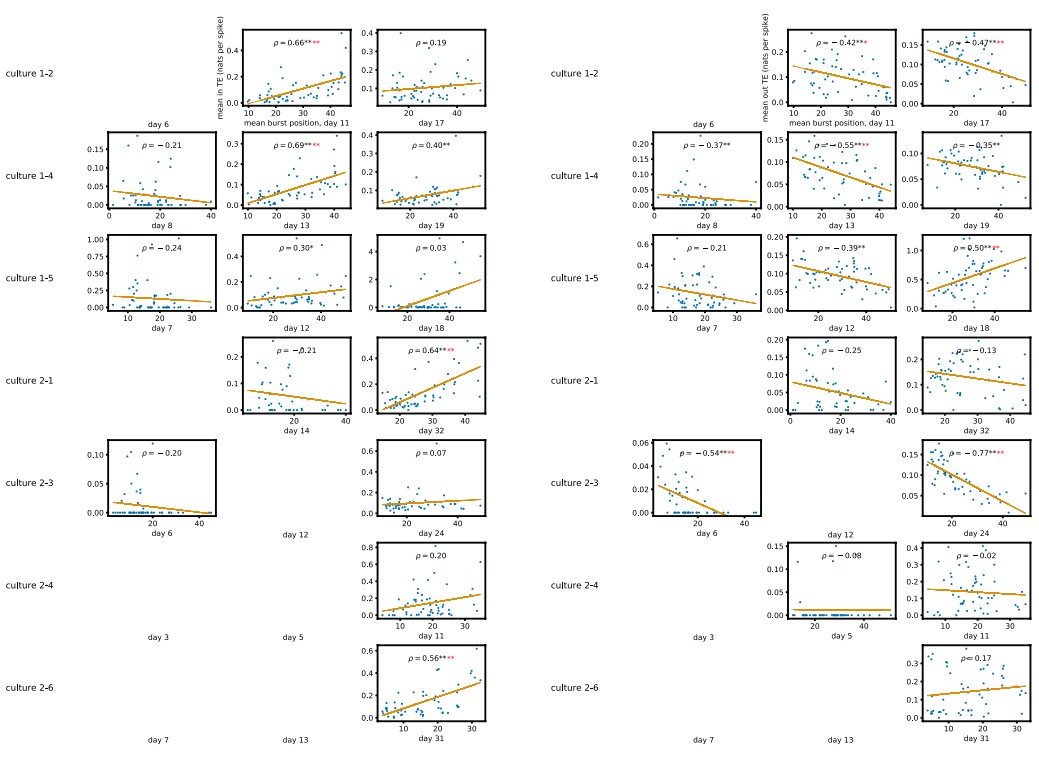

(a) Burst position vs TE in

(b) Burst position vs TE out

**Appendix 3—figure 7.** Identical plots to those in *Figure 5a and b*, but for the additional cultures. (**a**) Plots the mean burst position against the total incoming TE. (**b**) Plots the mean burst position against the total outgoing TE.

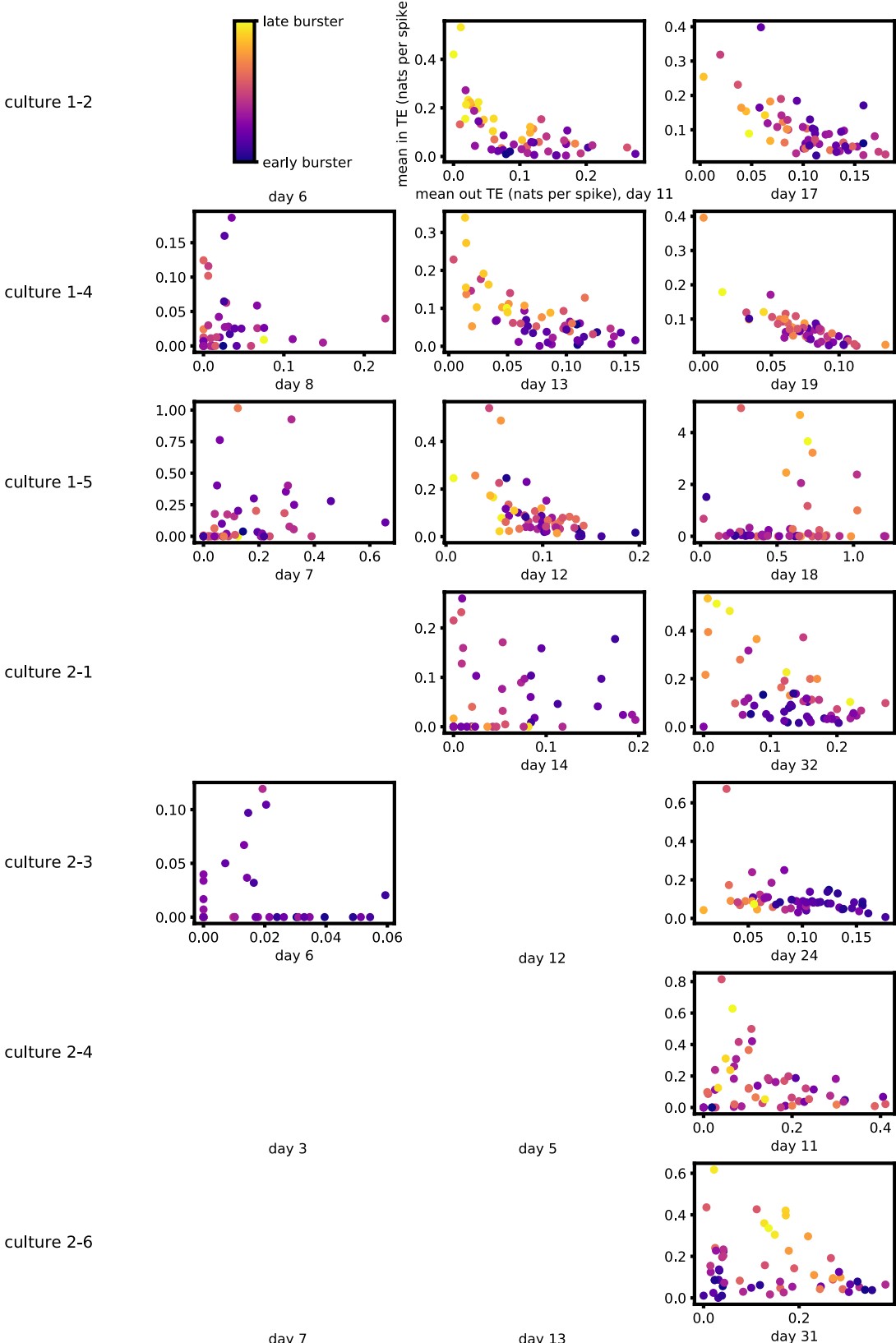

**Appendix 3—figure 8.** Identical plots to those in *Figure 5c*, but for the additional cultures.

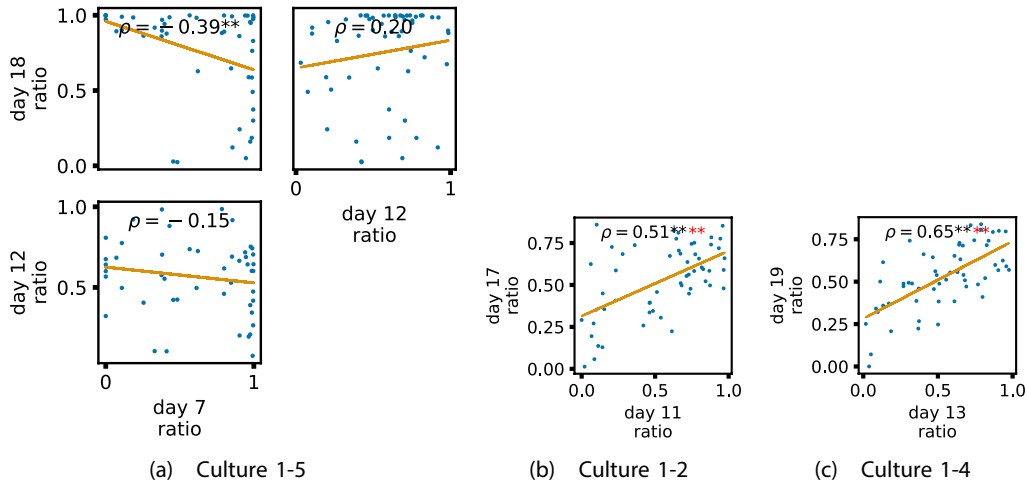

**Appendix 3—figure 9.** Identical plots to those in *Figure 6*, but for the additional cultures. (**a**) Contains plots for culture 1-5. (**b**) Contains plots for culture 1-2. (**c**) Contains plots for culture 1-4.

