## [Editor Report]

This work analyses how meaningful connections develop in the nervous system. The authors study the dissociated neuronal cultures and find that the information processing connections develop after 5–10 days. The direction of the information flow is influenced by neuronal bursting properties: the early bursting neurons emerge as sources and late bursting neurons become sinks in the information flow.

---

## [Decision Letter]

**Decision letter after peer review:**

Thank you for submitting your article "Early lock-in of structured and specialised information flows during neural development" for consideration by *eLife*. Your article has been reviewed by 2 peer reviewers, and the evaluation has been overseen by a Reviewing Editor and Michael Frank as the Senior Editor. The reviewers have opted to remain anonymous.

Essential revisions:

1) Please add results with additional methods as suggested by Reviewer 1.

2) Please expand the dataset to other publicly available data as suggested by Reviewer 2.

*Reviewer #1 (Recommendations for the authors):*

The submitted manuscript is well-structured and clearly written. In addition to the interesting and solid results presented in this paper by applying the continuous-time TE estimator on in vitro neural cultures, the reviewer is aware of the importance of developing accurate tools for the quantification of connectivity and information flows designed specifically for spike train data.

I suggest the acceptance of the manuscript for publication after the following comments are addressed:

A. Authors discovered interesting properties of developing neuronal networks showing that during bursting periods specific network nodes are engaged in specialized computational roles such as those of transmitters, receivers or mediators of information flow. Nevertheless, in order to investigate with full accuracy these specialized computational roles of neurons, it would be better to apply multivariate TE network inference (as the authors briefly discuss in lines 417-422), so as to elucidate the effects that nodes not involved in a bivariate analysis can have on the detection of these roles (especially that of mediator should be affected by spurious causality effects). Since the Authors have developed exactly this multivariate tool in their recent work, I wonder why such an approach has not been followed in this work, and how the presented results would change if a truly multivariate perspective to spike network analysis was undertaken.

B. The same dataset was analyzed by other researchers using various connectivity metrics, including also the same approach undertaken here. For instance, reference [64] employed the same continuous-time estimator of TE on the same data of dissociated neural cultures analyzed here, while other works [x, y] applied more standard metrics on these data, providing similar views about how these neuronal populations develop as a network across different stages of maturation. Reference and comparison to previous works is recommended to better place the present work in the context of the existing literature and to better understand the novelty and specificity of the results obtained.

[x] L. Minati et al., "Connectivity influences on nonlinear dynamics in weakly-synchronized networks: Insights from rossler systems, electronic chaotic oscillators, model and biological neurons," IEEE Access, vol. 7, pp. 174 793-174 821, 2019.

[y] J. H. Downes et al., "Emergence of a small-world functional network in cultured neurons," PLoS Comput. Biol, vol. 8, no. 5, p. e1002522, 2012.

C. The authors demonstrated the intrinsic early lock-in feature also in a model network of Izhikevich neurons, bringing this putative mechanism in a more general context. Still, it would be interesting to understand from a more detailed description how the different nature of biological and model neurons (MUA Vs. SUA) affects the underlying information flow.

D. Lines 88-89: Authors should provide the reason why and how they selected four cultures instead of analyzing all available 'overnight spontaneous dense' dataset of eleven cultures, and why the daytime spiking activity was discarded from the analysis ('daily spontaneous dense' dataset).

*Reviewer #2 (Recommendations for the authors):*

In this study the authors tackle the important problem of assessing information flow changes in developing neuronal cultures. They have previously developed a method that is the state of the art for information flow inference and apply it here to an existing dataset. But as mentioned in the public review, more data and analyses might be needed to validate their findings.

Main recommendation

I believe that this is an important work of wide interest for the systems and network neuroscience community. But the amount of data used is insufficient. Specially given that the publicly available dataset used is much richer. From the 2 used batches already, there's 5 and 6 cultures, with at least 15 different time points. Several more batches, with other firing patterns and densities are also present.

The work nicely demonstrates that neurons tend to assume the specialized computational roles of either transmitters, receivers or mediators of information flow, depending on burst position, i.e., early, middle and late bursters behave respectively as transmitters, mediators and receivers. A main strength of the work is the tool used for the analysis, i.e. a continuous-time estimator of the transfer entropy (TE) which was demonstrated in a recent work by the same authors to be far superior than the traditional discrete-time approach to TE estimation on neural data. The main weakness identified relies on a limited reference to previous literature analyzing the same publicly available data, and usage of insufficient dataset.

---

## [Author Response]

Essential revisions:1) Please add results with additional methods as suggested by Reviewer 1.

Response 1.1: Please note that reviewer 1 did not specifically ask us to use the additional methods (multivariate network inference), rather, they were asking us why we had not used this technique. We have provided a thorough explanation as to why we did not make use of multivariate TE for this paper below.

2) Please expand the dataset to other publicly available data as suggested by Reviewer 2.

Response 1.2: The analysis has been expanded to other publicly available data, as was suggested by Reviewer 2.

Reviewer #1 (Recommendations for the authors):The submitted manuscript is well-structured and clearly written. In addition to the interesting and solid results presented in this paper by applying the continuous-time TE estimator on in vitro neural cultures, the reviewer is aware of the importance of developing accurate tools for the quantification of connectivity and information flows designed specifically for spike train data.I suggest the acceptance of the manuscript for publication after the following comments are addressed:A. Authors discovered interesting properties of developing neuronal networks showing that during bursting periods specific network nodes are engaged in specialized computational roles such as those of transmitters, receivers or mediators of information flow. Nevertheless, in order to investigate with full accuracy these specialized computational roles of neurons, it would be better to apply multivariate TE network inference (as the authors briefly discuss in lines 417-422), so as to elucidate the effects that nodes not involved in a bivariate analysis can have on the detection of these roles (especially that of mediator should be affected by spurious causality effects). Since the Authors have developed exactly this multivariate tool in their recent work, I wonder why such an approach has not been followed in this work, and how the presented results would change if a truly multivariate perspective to spike network analysis was undertaken.

Response 2.2. We agree with the reviewer that multivariate TE network inference (in order to yield an effective network), using the conditional TE estimator for spike trains that we also developed, applied to this data set would be an interesting and valuable research contribution. However, there are several reasons why we have not included the use of the conditional TE with multivariate effective network inference in this work. In summary these include:

1. Our scope being establishing directed functional relationships as a first step in this paper;

2. Substantial validation work being required to establish linking our conditional TE estimator for spike trains into the multivariate TE inference algorithm; and

3. Our having reached the recommended word limit for Intro-Results-Discussion in *eLife*, leaving such large additions certainly out of scope.

These are outlined in the following:

1. Both effective and functional (as provided for here by our bivariate TE analysis) network analyses are widely accepted within neuroscience as providing valuable, *complementary*, insight. The perspective from which multivariate analyses are “better” is in terms of providing a more comprehensive understanding of the dynamics, and generally aligning more closely with underlying structure. Yet that does not detract from the complementary value of understanding functional and directed functional relationships, as should be noted from their pervasiveness in fMRI analysis relative to effective network inference. As we highlighted in the section of the Discussion the Reviewer mentions, it is our intention to apply a multivariate TE network inference analysis to this data set in the near future. This will then allow us to compare the results of the multivariate (effective) and bivariate (functional) analyses. However, such a comparison requires the functional analysis to have been conducted as a natural first step, which is what we are doing in the submitted manuscript.

2. When moving to multivariate TE analysis, the experimenter must ask which nodes to conditionon for a given source-target pair, or which set of nodes to measure the joint TE from. Attempting to measure conditional TE for a source-target pair conditioning on all other nodes for example (there would be 57 here) pushes well beyond the bounds of what has been shown to be empirically achievable for similar data sets even with this best of breed estimator (see Figure 8 of our previous work [4]). Instead, best practice is to greedily infer the set of parents for each target in an effective network inference algorithm with iteratively conditioned multivariate TE (see e.g. [3]). Yet before this greedy network inference algorithm should be applied to biological data in conjunction with our new TE estimator, scientific rigour dictates that the algorithm requires several subtle and non-trivial adaptations to be designed and introduced to accommodate the estimator (e.g. on how multiple and non-sequential past spikes of a source or conditional are included), and the combination of algorithm and estimator needs to be validated (on synthetic examples for which effective network can be expected to converge to the known underlying structure). This is a substantial amount of work and would form an individual paper on its own, mirroring our previous work validating the algorithm with standard estimators in [3] and [2].

3. We are already stretching the suggested word limit for *eLife* for Intro-Results-Discussion, and inclusion of another whole paper’s worth of material then is simply not achievable.

We have expanded the paragraph at the end of the discussion highlighting the inference of effective networks on this dataset as an exciting future direction.

B. The same dataset was analyzed by other researchers using various connectivity metrics, including also the same approach undertaken here. For instance, reference [64] employed the same continuous-time estimator of TE on the same data of dissociated neural cultures analyzed here, while other works [x, y] applied more standard metrics on these data, providing similar views about how these neuronal populations develop as a network across different stages of maturation. Reference and comparison to previous works is recommended to better place the present work in the context of the existing literature and to better understand the novelty and specificity of the results obtained.[x] L. Minati et al., "Connectivity influences on nonlinear dynamics in weakly-synchronized networks: Insights from rossler systems, electronic chaotic oscillators, model and biological neurons," IEEE Access, vol. 7, pp. 174 793-174 821, 2019.[y] J. H. Downes et al., "Emergence of a small-world functional network in cultured neurons," PLoS Comput. Biol, vol. 8, no. 5, p. e1002522, 2012.

Response 2.3: We agree with the reviewer that adding more discussion around previous work on this dataset will improve the paper.

We have added a new paragraph to the discussion, which addresses this previous work. It is now the second last paragraph of the discussion in the revised submission.

C. The authors demonstrated the intrinsic early lock-in feature also in a model network of Izhikevich neurons, bringing this putative mechanism in a more general context. Still, it would be interesting to understand from a more detailed description how the different nature of biological and model neurons (MUA Vs. SUA) affects the underlying information flow.

Response 2.4: We agree that it would be an interesting direction for future research to interrogate how aggregating single unit activity into multi-unit activity would change the estimated information flows. This would require incorporating a substantially more complex spatial model to extract MUA from, which conflicts with our goal of demonstrating corresponding early lock-ins and computational roles from a very simple model. As such, instead we have added a more detailed discussion as requested:

We have added a brief discussion to the 6th paragraph of the discussion, highlighting this potential future research direction.

D. Lines 88-89: Authors should provide the reason why and how they selected four cultures instead of analyzing all available 'overnight spontaneous dense' dataset of eleven cultures, and why the daytime spiking activity was discarded from the analysis ('daily spontaneous dense' dataset).

Response 2.5: As discussed in section ‘Cell culture data’ (lines 436 to 440 of the original submission), the daytime recordings were substantially shorter (30 minutes) than the overnight recordings. As information-theoretic estimators are incredibly data hungry, these were far less suitable for our proposed analysis than the longer overnight recordings. Of the overnight recordings then, 3 of the four cultures (1-3, 2-2 and 2-5) were selected because they were the only cultures for which four overnight recordings were conducted on four occasions, making them the most suitable for our intention of investigating development over time. The fourth culture (1-1) was selected as a sample from those with three overnight recordings based on having the latest final recording day that was still within a week of the penultimate recording day.

With that said, we do agree with this reviewer (and reviewer 2, see Response 3.7) that including more of the overnight recordings will improve the quality of the paper, and as such:

We have substantially expanded our analysis to include all 11 ‘overnight spontaneous dense’ cultures. The results of the extra runs are shown in the appendix and summarised and discussed in the revised text. Our high level conclusions remain as before, with some additional subtleties discussed.

Reviewer #2 (Recommendations for the authors):In this study the authors tackle the important problem of assessing information flow changes in developing neuronal cultures. They have previously developed a method that is the state of the art for information flow inference and apply it here to an existing dataset. But as mentioned in the public review, more data and analyses might be needed to validate their findings.Main recommendationI believe that this is an important work of wide interest for the systems and network neuroscience community. But the amount of data used is insufficient. Specially given that the publicly available dataset used is much richer. From the 2 used batches already, there's 5 and 6 cultures, with at least 15 different time points. Several more batches, with other firing patterns and densities are also present.

Response 3.7: We agree with the reviewer that adding more data will improve this contribution and we thank them for this suggestion (akin to that of reviewer 1 in Response 2.5). We will highlight, however, that it is not feasible to add in the daily recordings from this dataset. These recordings are all only 30 minutes long and, as such, they contain insufficient data for the application of informationtheoretic estimators. Instead:

We have expanded our analysis to include all 11 ‘overnight spontaneous dense’ cultures. The results of the extra runs are shown in the appendix and summarised and discussed in the revised text. As per Response 2.5, our high level conclusions remain as before, with some additional subtleties discussed.

The work nicely demonstrates that neurons tend to assume the specialized computational roles of either transmitters, receivers or mediators of information flow, depending on burst position, i.e., early, middle and late bursters behave respectively as transmitters, mediators and receivers. A main strength of the work is the tool used for the analysis, i.e. a continuous-time estimator of the transfer entropy (TE) which was demonstrated in a recent work by the same authors to be far superior than the traditional discrete-time approach to TE estimation on neural data. The main weakness identified relies on a limited reference to previous literature analyzing the same publicly available data, and usage of insufficient dataset.

Response 3.8: Please see Response 3.7 for a discussion of the extra dataset that has been added to the paper.

We agree with the reviewer that adding more discussion around previous work on this dataset will improve the paper (as per Response 2.3):

We have added a new paragraph to the discussion, which addresses this previous work. It is the second last paragraph of the revised submission.

References

Jerome L Myers, Arnold D Well, and Robert F Lorch Jr. *Research design and statistical analysis*. Routledge, 2013.Leonardo Novelli and Joseph T Lizier. Inferring network properties from time series using transfer entropy and mutual information: Validation of multivariate versus bivariate approaches. *Network Neuroscience*, 5(2):373–404, 2021.Leonardo Novelli, Patricia Wollstadt, Pedro Mediano, Michael Wibral, and Joseph T Lizier. Largescale directed network inference with multivariate transfer entropy and hierarchical statistical testing. *Network Neuroscience*, 3(3):827–847, 2019.David P Shorten, Richard E Spinney, and Joseph T Lizier. Estimating transfer entropy in continuous time between neural spike trains or other event-based data. *PLOS Computational Biology*, 17(4):e1008054, 2021.